# SAMPLE MORE TO THINK LESS: GROUP FILTERED POLICY OPTIMIZATION FOR CONCISE REASONING

**Vaishnavi Shrivastava**[m]    **Ahmed Awadallah**[m]    **Vidhisha Balachandran**[m]
**Shivam Garg**[m]    **Harkirat Behl**[m]    **Dimitris Papailiopoulos**[m,ω]

[m]Microsoft Research
[ω]University of Wisconsin-Madison

## ABSTRACT

Large language models trained with reinforcement learning on verifiable rewards often inflate response length—trading brevity for accuracy. While longer reasoning can help on hard problems, many extra tokens are filler: verbose text making little progress. We introduce GFPO (*Group Filtered Policy Optimization*), which curbs this length explosion by sampling larger groups per problem and only training on responses filtered by (1) length and (2) token efficiency (reward per token). By sampling *more* during training time, GFPO teaches models to think *less* at inference time. On Phi-4-reasoning, GFPO cuts GRPO's length inflation by up to 85% across STEM and coding benchmarks (AIME 24/25, GPQA, Omni-MATH, LiveCodeBench) while preserving accuracy. We find that GFPO also outperforms Dr. GRPO in both accuracy and length reduction and generalizes across model sizes and families. We further propose Adaptive Difficulty GFPO, which allocates more training exploration to harder problems, yielding better efficiency-accuracy trade-offs on challenging questions. With only a 7% increase in training time, GFPO reduces end-to-end latency by ∼30%, cutting response time on hard queries by 90 seconds. GFPO trades modest training-time increases for lasting gains in inference—an effective recipe for efficient reasoning.

**GRPO**

$$\widehat{A}_{i,t} = \frac{R(q,o_i) - \text{mean}\{R(q,o_1),...,R(q,o_G)\}}{\text{std}\{R(q,o_1),...,R(q,o_G)\}}$$

$$\mathcal{L}_{GRPO_{i,t}} = \min\left[\frac{\pi_\theta(o_{i,t} \mid q, o_{i,<t})}{\pi_{\theta_{old}}(o_{i,t} \mid q, o_{i,<t})}\widehat{A}_{i,t}, \text{clip}\left(\frac{\pi_\theta(o_{i,t} \mid q, o_{i,<t})}{\pi_{\theta_{old}}(o_{i,t} \mid q, o_{i,<t})}, 1-\varepsilon, 1+\varepsilon\right)\widehat{A}_{i,t}\right]$$

**GFPO**

$$\mathcal{S} = \text{REJECTIONSAMPLE}(\mathcal{G}, k, \text{metric}), \; m_i = \mathbb{I}_{\{i \in \mathcal{S}\}}$$

$$\widehat{A}_{i,t}^{(m)} = \frac{R(q,o_i) - \text{mean}\left\{R(q,o_{s_1}),...,R(q,o_{s_k})\right\}}{\text{std}\left\{R(q,o_{s_1}),...,R(q,o_{s_k})\right\}}m_i$$

$$\mathcal{L}_{GFPO_{i,t}} = \min\left[\frac{\pi_\theta(o_{i,t}|q,o_{i,<t})}{\pi_{\theta_{old}}(o_{i,t}|q,o_{i,<t})}\widehat{A}_{i,t}^{(m)}, \text{clip}\left(\frac{\pi_\theta(o_{i,t}|q,o_{i,<t})}{\pi_{\theta_{old}}(o_{i,t}|q,o_{i,<t})}, 1-\varepsilon, 1+\varepsilon\right)\widehat{A}_{i,t}^{(m)}\right]$$

Figure 1: **Left:** GFPO introduces simple yet powerful modifications to GRPO: sample more responses during training (↑ $G$), rank them by a target attribute (e.g., length, token efficiency), and learn only from the top-$k$—setting the advantages of the rest to zero. This selective learning functions as implicit reward shaping, steering the policy toward desired behaviors. **Right:** When optimizing for length or token efficiency, GFPO curbs GRPO's length inflation—letting the model *think less* at inference-time by *sampling more* at training-time—while maintaining its core reasoning capabilities.

## 1 INTRODUCTION

Reinforcement learning from verifier rewards (RLVR) methods such as GRPO (Shao et al., 2024) and PPO (Schulman et al., 2017) have been pivotal for test-time scaling—enabling models like O3 (OpenAI, 2025) and DeepSeek-R1 (Guo et al., 2025) to "think longer" and achieve state-of-the-art results on challenging reasoning tasks such as AIME and IMO. Yet longer chains are not always better: prior work shows that long responses don't correlate with correct answers—shorter responses can even be more accurate. For instance, Balachandran et al. (2025) report that DeepSeek-R1 produces responses nearly 5× longer than Claude 3.7 Sonnet on AIME 25 with no accuracy gain, while Hassid et al. (2025) find that QwQ-32B's shortest responses outperform random ones by 2% while using 31% fewer tokens.

One may suspect that longer responses simply reflect harder problems. However, by comparing correct and incorrect responses to the same AIME 25 questions with Phi-4-reasoning-plus (Abdin et al., 2025), we find the opposite: in 72% of cases, the longer responses are more likely to be wrong. This suggests that verbosity is not just a byproduct of difficulty but a distinct failure mode.

Works such as Dr. GRPO (Liu et al., 2025) and DAPO (Yu et al., 2025) apply token-level normalization to address this failure mode. However, even with these methods, response length for Phi-4-reasoning-plus balloons from 4k to 14k tokens within 100 GRPO steps. We hypothesize that while token-level normalization penalizes long incorrect outputs, it also amplifies rewards for long correct ones—reinforcing verbosity in models already SFTed for step-by-step reasoning (Abdin et al., 2025; Guo et al., 2025).

Motivated by these observations, our goal is to train efficient reasoning models: ones that preserve GRPO's accuracy while producing far shorter reasoning chains. Our contributions are as follows:

- **GFPO (Group Filtered Policy Optimization)**: A variant of GRPO that samples larger groups of candidate chains to increase exposure to desirable outputs, filters them based on a target metric, and learns only from the filtered subset. GFPO optimized for response length reduces GRPO's length inflation by 46–71% across AIME 24/25, GPQA, Omni-MATH, and LiveCodeBench, with no loss in accuracy (§4.1).
- **Token Efficiency** (§4.2): Defined as the ratio of reward to response length—allows longer chains when justified by higher rewards. GFPO with this metric cuts length inflation by 71–85%.
- **Adaptive Difficulty GFPO** (§4.3): A dynamic variant of GFPO that allocates more exploration to hard problems using unsupervised difficulty estimates, striking a better balance between efficiency and accuracy.
- **Out-of-Distribution Generalization** (§4.4): We demonstrate that GFPO preserves accuracy while curbing response length even for out-of-distribution tasks.

Our experiments further show that GFPO improves upon Dr. GRPO, delivering both higher accuracy and shorter reasoning chains (§4.6). GFPO also generalizes across model families and sizes—including Phi-4-Reasoning and the DeepSeek-R1-Distill Qwen and Llama models at 7B, 8B, and 14B scales—cutting response lengths while consistently preserving GRPO-level accuracy (§4.7).

Our best GFPO variant increases training time by only 7%, yet reduces inference latency by nearly 30% for hard problems compared to GRPO. This translates to long responses arriving ∼90 seconds faster for users—an immediate and substantial benefit (§5.3). GFPO provides a favorable train-test trade-off, delivering materially lower inference latency with only marginal additional training cost.

## 2 GROUP FILTERED POLICY OPTIMIZATION

Group Relative Policy Optimization (GRPO; Shao et al. (2024)) simplifies Proximal Policy Optimization (PPO; Schulman et al. (2017)) by removing the value model and instead using the average reward of sampled responses as a baseline, while retaining PPO's clipped surrogate objective.

We propose *Group Filtered Policy Optimization* (GFPO), a simple yet effective method for targeted policy optimization of desirable response properties. GFPO samples a larger group of candidate responses per question, broadening the response pool to include more candidates with desirable traits,

and then explicitly filters for these traits when computing the policy gradient. While it may seem natural to directly encode desirable attributes such as brevity or informativeness into the scalar reward, doing so for multiple traits can be challenging, especially when correctness must already be captured.

Data filtration instead serves as an implicit, flexible form of reward shaping—akin to iterative self-improvement methods that use selective sampling to amplify specific model behaviors (Zelikman et al., 2022). After this explicit filtering step isolates the preferred responses, standard rewards are then used solely to compute relative advantages within the selected group. Thus, GFPO optimizes for multiple desirable properties (e.g., length and accuracy) simultaneously, without requiring complex reward engineering. Since our goal is to reduce the response length inflation in RL, we focus on using GFPO to optimize for shorter responses while matching GRPO's accuracy.

Given a question $q$, we sample a large set of responses $\mathcal{G} = \{o_1, \ldots, o_G\}$ from the current policy. Rather than training equally on all responses, GFPO applies a selection step based on a user-specified metric to filter a subset of size $k$ of the most desirable responses to train on. We compute a metric score for each response and sort accordingly, selecting the top-$k$ responses to form the retained subset $\mathcal{S} \subseteq \mathcal{G}$. We define a binary mask $m \in \{0, 1\}^G$, where $m_i = 1$ indicates a selected response and $m_i = 0$ indicates a rejected response.

Formally, we define the GFPO objective[1] as:

$$\mathcal{J}_{\text{GFPO}}(\theta) = \mathbb{E}_{q \sim P(Q), \{o_i\}_{i=1}^G \sim \pi_{\theta_{\text{old}}}(O|q)} \frac{1}{\sum_{i=1}^G |o_i|} \sum_{i=1}^G \sum_{t=1}^{|o_i|} \min\left(r_{i,t} \widehat{A}_{i,t}^{(m)}, \text{clip}(r_{i,t}, 1-\varepsilon, 1+\varepsilon) \widehat{A}_{i,t}^{(m)}\right)$$

$$-\beta \, \mathcal{D}_{KL}\left(\pi_\theta \,\|\, \pi_{\theta_{\text{old}}}\right) + \gamma \, \text{Entropy}(\pi_\theta) \tag{1}$$

where

$$\mathcal{S}, m = \text{REJECTIONSAMPLE}(\mathcal{G}, k, \text{metric}, \text{order}), \; m_i = \mathbb{I}_{\{i \in \mathcal{S}\}}$$

$$\widehat{A}_{i,t}^{(m)} = \frac{R(q, o_i) - \frac{1}{k}\sum_{j \in S} R(q, o_j)}{\sqrt{\frac{1}{k}\sum_{j \in S}\left(R(q, o_j) - \frac{1}{k}\sum_{p \in S} R(q, o_p)\right)^2}} m_i, \qquad r_{i,t} = \frac{\pi_\theta(o_{i,t} \,|\, q, o_{i,<t})}{\pi_{\theta_{\text{old}}}(o_{i,t} \,|\, q, o_{i,<t})}$$

and $\beta \, \mathcal{D}_{KL}\left(\pi_\theta \,\|\, \pi_{\theta_{\text{old}}}\right)$ denotes the KL penalty.

We normalize the advantages for responses in the selected subset $\mathcal{S}$ using the mean and standard deviation of the response-level rewards in $\mathcal{S}$. This enables meaningful comparisons among responses already exhibiting the desired property, ensuring GFPO prioritizes the highest-reward responses within the filtered subset. Responses not in $\mathcal{S}$ receive zero advantage, effectively excluding them from influencing policy updates. Thus, GFPO's primary intervention is at the level of advantage estimation, making it compatible with any GRPO variant such as DAPO (Yu et al., 2025), Dr. GRPO (Liu et al., 2025), or GRPO with the Dual-Clip PPO loss (Ye et al., 2020).

While GFPO is general-purpose and can accommodate various scoring metrics, our experiments specifically leverage metrics aimed at reducing response length inflation:

- **Response Length**: Training on short responses directly encourages brevity.
- **Token Efficiency** (reward/length): Training on highly token-efficient responses encourages succinctness, but still allows longer responses if sufficiently "justified" by proportionately higher rewards.

Other metrics—such as factuality, diversity, or external quality scores—could also be integrated into GFPO to optimize different attributes of interest.

**2.1 Adaptive Difficulty GFPO.** We introduce Adaptive Difficulty GFPO which allocates more training signal to harder questions. At each step, we estimate difficulty from the average reward of sampled responses—lower averages indicate higher difficulty.

To scale the number of retained responses $k$, we maintain a streaming summary of prompt difficulties using a lightweight t-digest, which approximates quartiles over past rewards. New questions are

---

[1]Note we use the DAPO token-level loss aggregation for both GFPO and GRPO which is the default choice in `verl`. We employ a slightly modified version of the clipped surrogate policy gradient loss introduced in prior work (Li et al., 2025), which reduces training instabilities caused by negative advantages and large policy ratios.

bucketed into difficulty levels, and assigned $k = 4$ (easy), $k = 6$ (medium), or $k = 8$ (hard/very hard) of 16 samples. During a short warmup period, all questions use $k = 8$ to avoid unstable estimates. The number of buckets and $k$ per bucket are hyperparameters. See Appendix A.6 for more details.

Our method sharpens filtering on easy prompts while encouraging exploration on harder ones. To our knowledge, this is the first RLVR method that adapts group size based on question difficulty.

## 3 SETUP

**Model.** We build on Phi-4-reasoning (Abdin et al., 2025), a 14B-parameter Phi-4 model (Abdin et al., 2024) extensively SFTed on synthetic o3-mini reasoning traces in STEM, but never trained with RL. We refer to Phi-4-reasoning as the SFT baseline. To assess GFPO's generalization across model families and scales, we also evaluate GFPO on DeepSeek-R1-Distill-Qwen-7B, DeepSeek-R1-Distill-Llama-8B, and DeepSeek-R1-Distill-Qwen-14B.

**Baseline.** We compare our GFPO trained models with Phi-4-reasoning-plus (Abdin et al., 2025), which is trained with GRPO and DAPO's token-level loss aggregation on top of the Phi-4-reasoning model. We refer to this as the GRPO baseline. We match the training setup of Phi-4-reasoning, and use a slightly modified clipped surrogate objective for training stability (§2). For the DeepSeek-R1-Distill models, we likewise train GRPO baselines as described below. We also compare GFPO with Dr. GRPO (Liu et al., 2025), a prominent alternative for token-efficient reasoning.

**Dataset.** RL training uses 72k math problems from the same corpus as Abdin et al. (2025). With 100 training steps and batch size 64, models see only 6.4k problems—identical to the GRPO baseline.

**Reward Function.** We adopt the GRPO baseline reward: a weighted sum of (i) a length-aware binary accuracy reward $R_{\text{acc}}$ and (ii) a 5-gram repetition penalty $R_{\text{rep}}$:

$$R = w_{\text{acc}}\text{LENGTHSCALE}(R_{\text{acc}}) + w_{\text{rep}}R_{\text{rep}}, \quad R \in [-1, 1]. \tag{2}$$

Accuracy is $0/1$ based on extracted final answers, with GPT-4o fallback if regex extraction fails. Formatting violations receive the minimum reward. Long correct responses are penalized by cosine reward scaling. However, this length penalty is insufficient to prevent GRPO's length inflation.

**Training Configuration.** For the 14B models, we train with `verl` (Sheng et al., 2024) on 32 H100s, with global batch size of 64, for 100 steps. We use Adam with learning rate $1 \times 10^{-7}$, cosine warmup (10 steps), temperature $T = 1.0$, KL regularization ($\beta = 0.001$), and entropy coefficient ($\gamma = 0.001$). Models are trained with a 32k context, reserving 1k tokens for the prompt. GRPO uses group size $G = 8$. GFPO increases $G \in \{8, 16, 24\}$ to expose the model to more candidates, but retains only $k \leq 8$ responses for policy gradients, ensuring a fair comparison. For the 7B and 8B models, we train with 16 H100s, with a learning rate of $1 \times 10^{-6}$, and 16k context. All other hyperparameters match the 14B training configuration. See Appendix A.7 for details on training dynamics.

**Evaluation.** We evaluate on: AIME 25/24 (AIME, 2025; 2024) (32 samples), GPQA (Rein et al., 2024) (5 samples), Omni-MATH (Gao et al., 2025) (1 sample), and LiveCodeBench (8/24–1/25) (Jain et al., 2024) (3 samples). Responses are sampled at $T = 0.8$, top-p=0.95 with 32k max length, with a maximum of 1k prompt tokens. For the DeepSeek-R1-Distill-Qwen models, we evaluate at $T = 0.6$ with 16k max length for the 7B model. Final answers are extracted via regex, with GPT-4o fallback. LiveCodeBench tests OOD generalization to code, which is unseen during RL training.

We report **pass@1 accuracy**, raw response length $L$, and **excess length reduction (ELR)**:

$$ELR = \frac{L_{\text{GRPO}} - L_{\text{GFPO}}}{L_{\text{GRPO}} - L_{\text{SFT}}}. \tag{3}$$

Statistical significance is tested using the Wilcoxon signed-rank test (Wilcoxon, 1992).

## 4 RESULTS

**4.1 GFPO Reduces Length by Sampling More and Retaining Less.** An initial question is whether rejection sampling alone, without increasing sampled responses, suffices to shorten reasoning chains. To examine this, we evaluate Shortest 6/8 GFPO, which retains the six shortest responses from a

| | AIME 25 | AIME 24 | GPQA | Omni-MATH | LiveCode Bench | Average | | |
|---|---|---|---|---|---|---|---|---|
| | % Len Inf ($\downarrow$) | % Len Inf ($\downarrow$) | % Len Inf ($\downarrow$) | % Len Inf ($\downarrow$) | % Len Inf ($\downarrow$) | Acc | Len | % Len Inf ($\downarrow$) |
| SFT | N/A | N/A | N/A | N/A | N/A | 69.2 | 9.5k | N/A |
| GRPO | 0.0 | 0.0 | 0.0 | 0.0 | 0.0 | 72.1 | 13k | 0.0 |
| Dr. GRPO | 43.6 | 48.5 | 65.1 | 71.6 | 7.2 | 70.1 | 11.5k | 47.2 |
| 6 of 8 | 1.8 | 9.5 | 11.5 | -5.5 | 7.0 | 72.7 | 12.9k | 4.8 |
| 8 of 16 | 23.8 | 33.0 | 23.7 | 31.5 | 36.5 | **73.4** | 12k | 29.7 |
| 6 of 16 | 25.6 | 35.6 | 38.8 | 43.7 | **37.2** | 72.3 | 11.8k | 36.2 |
| 4 of 16 | **38.0** | **46.8** | **45.7** | 47.3 | 43.2 | 72.0 | **11.5k** | **44.2** |
| 8 of 24 | **54.4** | 52.7 | 52.2 | 51.9 | **59.4** | 71.7 | 11.1k | 54.1 |
| 6 of 24 | 41.0 | 44.9 | 48.6 | 58.2 | 42.7 | 72.2 | 11.4k | 47.1 |
| 4 of 24 | 46.1 | **59.8** | 57.3 | 71.0 | 57.0 | 72.3 | **11k** | **58.2** |
| Token Eff. | **70.9** | **84.6** | **79.7** | **82.6** | **79.7** | 71.7 | **10.2k** | **79.5** |
| Adaptive Diff. | 50.8 | 52.9 | 41.7 | 35.1 | 49.4 | **72.9** | 11.4k | 46.0 |

Table 1: **Pass@1 Accuracy, Response Lengths, and Length Inflation Reduction.** Across all benchmarks, GFPO cuts length inflation while matching GRPO accuracy (no significant difference under Wilcoxon signed-rank test). Sampling more responses is key and lowering $k/G$ effectively controls length. Token Efficiency delivers the largest reduction in length inflation (79.5%) at GRPO-level accuracy, and Adaptive Difficulty outperforms shortest $k/G$ at equal compute. On LiveCodeBench (OOD coding), GRPO lengthens chains without accuracy gains, whereas GFPO shortens them and sometimes improves accuracy (e.g., 8/16, 4/24). GFPO also outperforms Dr. GRPO, with higher accuracy and larger excess-length reductions. Pass@1 accuracy uses 32 (AIME-25/24), 5 (GPQA), 1 (Omni-MATH), and 3 (LCB) samples. See Table 4 for per-dataset response lengths and pass@1.

group of eight. While accuracy remains comparable to GRPO on AIME 25/24, GPQA, and Omni-MATH, length reductions are minimal (1.8–11.5%) and even negative on Omni-MATH (+5.5%). This indicates that subsampling within small response groups offers little efficiency benefit.

Substantial improvements emerge once the group size is increased. With Shortest 8/16 GFPO, which filters the shortest half of 16 candidates, excess length is reduced by 24–37% across benchmarks without statistically significant accuracy loss. Further decreasing the number of retained responses strengthens this effect: Shortest 6/16 and 4/16 achieve an additional 2–22% reduction relative to 8/16. Scaling the group size amplifies these gains—for example, increasing from 8/16 to 8/24 yields 20–30% additional reduction, while 4/24 results in an additional 4% reduction over 8/24 (Table 1).

Taken together, these results indicate that the decisive factor is the retention fraction $k/G$ (Figure 6). Decreasing this fraction—either by reducing $k$ or increasing $G$—consistently shortens reasoning chains. 4/16 and 6/24 both retain 25% of responses, and their length reductions are nearly identical—confirming that $k/G$ is the key factor. Sampling from a larger group offers only a slight additional benefit, as seen with 6/24. Beyond a point, however, returns diminish: moving from 8/24 to 4/24 yields only marginal additional gains. The strongest reductions are observed at retention fractions of about 25–33%. We offer additional guidance for practitioners on tuning and scaling $k$ and $G$ in Appendix A.5.

**4.2 Reinforcing Token Efficiency.** Reducing the $k/G$ ratio eventually stalls learning—failing to deliver meaningfully shorter chains beyond a certain group size. To break this ceiling, we introduce Token Efficiency GFPO, which ranks responses by reward-per-token ($R_i/|o_i|$)—favoring longer chains only when their rewards justify the added cost. Token Efficiency GFPO filters for high reward-per-token responses—typically short correct chains, plus some long correct and long incorrect ones. Within this set, short correct chains receive the strongest positive gradients, long correct ones are modestly penalized, and long incorrect ones are sharply cut back, providing more direct length control than shortest-$k$, which relies on the KL penalty to implicitly suppress late-token probabilities.

With $k = 8, G = 16$, this method yields the largest length reductions across tasks—70.9% (AIME 25), 84.6% (AIME 24), 79.7% (GPQA), 82.6% (Omni-MATH), and 79.7% (Live-

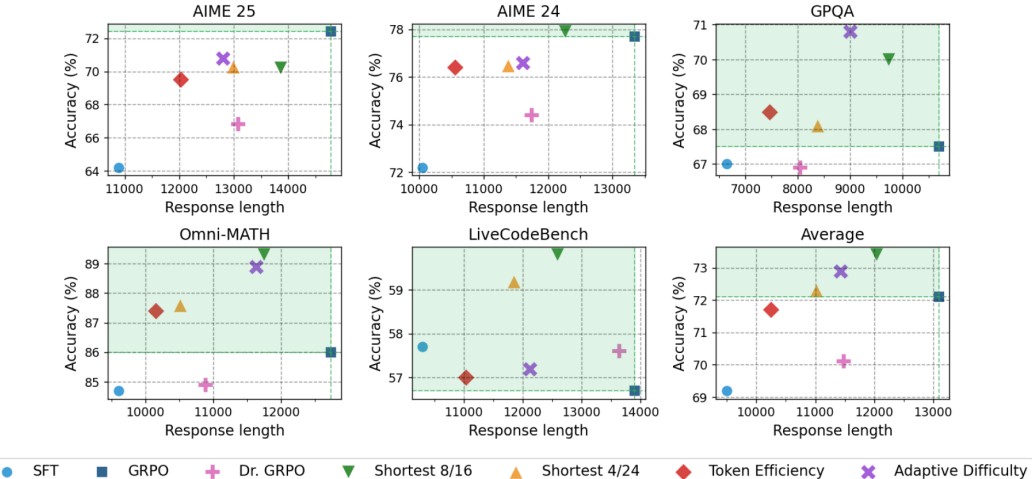

Figure 2: **Pareto Trade-off Between Accuracy and Response Length.** For all benchmarks except AIME 25, at least one GFPO variant strictly dominates GRPO—achieving both higher accuracy and shorter responses (green region above and to the left of GRPO). For AIME 25, GRPO attains the highest accuracy, but several GFPO variants, while taking non-significant accuracy dips, remain Pareto-optimal because their responses are shorter, and no other method is simultaneously more accurate and more concise. On average, Shortest 4/24, Adaptive Difficulty, and Shortest 8/16 are strictly Pareto-superior to GRPO with Token Efficiency close behind. Dr. GRPO generally falls outside the Pareto frontier–yielding lower accuracy and longer responses than GFPO.

CodeBench)—outperforming shortest-$k$ at similar or smaller $G$ (Table 1). These gains come with small, non-significant drops in accuracy. Still, Token Efficiency GFPO consistently delivers the sharpest token savings without compromising accuracy—showing reward-per-token to be a powerful proxy for concise reasoning.

**4.3 Adaptive Difficulty GFPO.** Beyond improved rejection metrics, we introduce Adaptive Difficulty GFPO, which varies the retained group size $k$ by question difficulty, allocating more training to harder problems. We estimate difficulty via per-question average reward, compute quartiles each step, and assign questions to four buckets: very hard (0–25%), hard (25–50%), medium (50–75%), and easy (75–100%). From $G=16$ samples, we retain 8, 8, 6, and 4 shortest responses in these buckets, respectively, giving an average $k=6.5$ and making Shortest 6/16 GFPO a natural baseline.

Adaptive Difficulty GFPO achieves stronger excess length reductions than Shortest 6/16 on AIME 25 (51% vs. 26%), AIME 24 (53% vs. 36%), GPQA (42% vs. 39%), and LiveCodeBench (46% vs 36%) though Shortest 6/16 is more effective on Omni-MATH (44% vs. 35%). Even against the more aggressive Shortest 4/16, it performs better on AIME 25 (51% vs. 38%), AIME 24 (53% vs. 47%), and LiveCodeBench (49% vs 43%) (Table 1). It also delivers the highest accuracy on GPQA (71%) and on the hardest AIME 25 quartile (27%) compared to GRPO and other GFPO variants (Figure 3b).

**4.4 Out-of-Distribution Effects of GFPO.** Our RL training recipe is geared towards enhancing mathematical reasoning performance. To investigate potential adverse effects of GFPO's bias toward shorter responses, we assess out-of-distribution generalization on the LiveCodeBench coding benchmark. Note that coding is not a part of our RL training set. GRPO inflates response length even out-of-distribution—outputs grow from 10.3k tokens (SFT) to 13.9k, while accuracy stagnates (57% vs. 58%) (Table 4). This verbosity is undesirable, especially without accuracy gains. GFPO counters this: Token Efficiency reduces excess length by 80%, and Shortest 8/24 trims 57% while modestly improving accuracy to 59% (vs. 58% SFT, 57% GRPO). GFPO not only reins in unnecessary length but can also enhance out-of-distribution generalization.

**4.5 Accuracy-Length Pareto Comparison.** Figure 2 shows the accuracy–length frontier. On four of five benchmarks, at least one GFPO variant is strictly *Pareto-superior* to GRPO (green region), demonstrating that GFPO can yield both shorter and more accurate answers. Even on AIME 25, where GRPO is slightly more accurate, GFPO variants remain on the Pareto front by offering meaningful length reductions without significant accuracy loss. Aggregated results (bottom-right)

| | | AIME 25 | | | AIME 24 | | | GPQA | | |
|---|---|---|---|---|---|---|---|---|---|---|
| | | Acc | Len | % Len Inf (↓) | Acc | Len | % Len Inf (↓) | Acc | Len | % Len Inf (↓) |
| Deepseek-R1 Distill Qwen 7B | SFT | 37.0 | 11.4k | N/A | 49.5 | 10.7k | N/A | 47.4 | 7.5k | N/A |
| | GRPO | **40.2** | 13.3k | 0 | 51.5 | 11.9k | 0 | **48.4** | 8.3k | 0 |
| | 8 of 16 GFPO | 39.1 | **11.9k** | 63 | **52.4** | **11k** | 39.6 | 48 | **7.9k** | 48.9 |
| DeepSeek-R1 Distill Llama 8B | SFT | 29.8 | 12.7k | N/A | 44.3 | 12.1k | N/A | 45.7 | 7.5k | N/A |
| | GRPO | 33.0 | **10.4k** | 0 | 47.9 | 10.3k | 0 | 45.5 | 7.6k | 0 |
| | 8 of 16 GFPO | **33.3** | **10.4k** | Undef | **48.5** | **10.0k** | Undef | 45.7 | **7.3k** | **319** |
| Deepseek-R1 Distill Qwen 14B | SFT | 48.2 | 12.3k | N/A | 64 | 10.8k | N/A | 56.4 | 7k | N/A |
| | GRPO | 51.9 | 16.1k | 0 | **69.3** | 13.9k | 0 | 56.8 | 8.8k | 0 |
| | 8 of 16 GFPO | **52.0** | **13.7k** | 61.9 | 68.5 | **12.5k** | 44.9 | **57.4** | **8.4k** | 22.8 |

Table 2: **GFPO on DeepSeek-R1 Distill Models.** GFPO generalizes to DeepSeek-R1-Distill models of the Qwen and Llama families at the 7B, 8B, and 14B scales, consistently reducing excess response length while preserving GRPO-level accuracy. Notably for the R1-Distill Llama model, both GRPO and GFPO reduce length *below* the SFT model while improving accuracy for AIME. GFPO also shortens length for GPQA.

highlight Shortest 4/24, Adaptive Difficulty, and Shortest 8/16 as the most consistently concise and accurate, with Token Efficiency trailing in accuracy by a narrow margin.

**4.6 GFPO vs Dr. GRPO.** We compare GFPO with Dr. GRPO (Liu et al., 2025), which removes GRPO's per-response length normalization. Across all tasks, Dr. GRPO delivers lower accuracy and longer responses—GFPO improves accuracy by 1–3% on AIME 24, AIME 25, GPQA, and Omni-Math, with 10-70 % more excess length reduction than Dr. GRPO. (Table 1, Table 4).

This gap follows directly from Dr. GRPO's design: without length normalization, gradients scale with token count, so long trajectories dominate updates. Since our models are already SFTed on chain-of-thought and biased toward long traces, Dr. GRPO mainly learns to select among long trajectories rather than shift probability mass toward shorter ones. While this may help in cold-start RL, GFPO's explicit preference for concise, high-quality samples is better suited to reasoning models with SFT priors. We also observe less stable training under Dr. GRPO, with spikes in KL divergence, gradient norm, and entropy loss. Meanwhile, GFPO matches GRPO's stability (Figures 8, 9).

**4.7 Generalization Across Model Families and Sizes.** To evaluate cross-model generalization, we apply GFPO to the DeepSeek-R1-Distill Qwen and Llama families at 7B, 8B, and 14B scales (Table 2). On DeepSeek-R1-Distill-Qwen-7B, GFPO cuts length inflation by 63%, 39.6%, and 48.9% on AIME 25, AIME 24, and GPQA, respectively, while closely tracking GRPO's accuracy. The 14B model shows the same trend: GFPO reduces inflation by 61.9%, 44.9%, and 22.8% on the three benchmarks without compromising accuracy. For DeepSeek-R1-Distill-Llama-8B, surprisingly both GFPO and GRPO *shorten* responses below the SFT baseline on AIME 24 and 25 while still improving accuracy–with GFPO providing equal (AIME 25) or greater (AIME 24) length reduction. On GPQA, GRPO slightly lengthens responses, whereas GFPO shortens them. GFPO generalizes across model families and scales, suppressing length inflation while maintaining GRPO-level accuracy.

## 5 ANALYSIS

We analyze performance of GFPO models on AIME 2025 by measuring question difficulty as $1 - \text{SFT}$ accuracy, which captures how hard each problem is for the SFT model prior to RL. Problems are partitioned into quartiles (easy–very hard) to study how GFPO affects length and accuracy across difficulty and accuracy of long responses at fixed difficulty. We also examine which parts of responses GFPO trims, with qualitative comparisons to GRPO for AIME 25 and GPQA in Appendix A.2.

**5.1 Length and Accuracy Across Problem Difficulty.** On AIME 2025, response lengths grow steeply with problem difficulty—from roughly 4k tokens on easy questions to over 20k on very hard ones (Figure 7). GFPO consistently reduces this verbosity across all quartiles (Figure 3a). Token

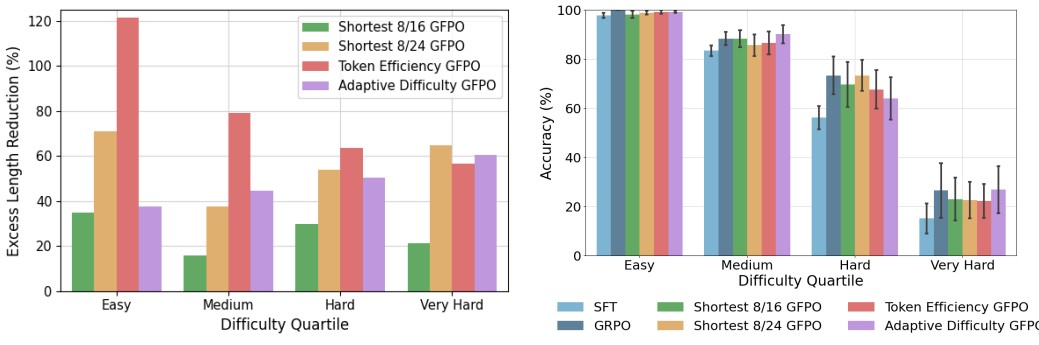

(a) **Excess Length Reduction Across Problem Difficulties.**

(b) **Accuracy Across Problem Difficulties.**

Figure 3: **Excess Length Reductions and Accuracy Across AIME 25 Problem Difficulties.** (**a**) GFPO reduces excess length across all difficulties. Token efficiency has the strongest overall reductions–with outputs more brief than the SFT model on easy questions. Shortest 8/24 has the best reductions on very hard questions. (**b**) Adaptive Difficulty and Shortest 8/24 have the best accuracies.

Efficiency GFPO delivers the largest overall reductions, exceeding 120% excess length reduction on easy problems—producing shorter outputs than the SFT baseline, while maintaining accuracy. Its impact diminishes on harder problems (56–79% reduction) because the token efficiency criterion permits longer responses when justified by higher rewards. Adaptive Difficulty GFPO follows the opposite trend: modest gains on easy questions (38%) but substantially stronger reductions on very hard ones (60%), effectively suppressing the "long tail" of overly verbose outputs. Shortest 8/24 also consistently surpasses Shortest 8/16, achieving the strongest reductions on very hard problems.

Accuracy patterns mirror these differences (Figure 3b). All methods perform near-perfectly on easy problems, while GRPO and GFPO both improve over the SFT baseline on harder ones. Token Efficiency's reductions come with small, statistically insignificant accuracy dips. Adaptive Difficulty, by contrast, matches or exceeds GRPO accuracy across easy, medium, and very hard questions (e.g., 90% vs. 88% on medium; 27% vs. 27% on very hard) while simultaneously reducing length by up to 60%. Its only shortcoming appears on "hard" questions, where filtering occasionally removes useful longer responses. This can be mitigated by increasing the group size: for example, Shortest 8/24 fully recovers GRPO's 73% accuracy on hard questions while producing substantially shorter outputs.

**5.2 Accuracy of Long Responses under GFPO.** Reasoning models often produce less accurate answers as response length grows, but this effect is entangled with problem difficulty—harder questions naturally elicit longer chains. To isolate verbosity, we fix difficulty and examine accuracy by response length on AIME 2025. Using SFT per-question accuracy as a difficulty proxy, we partition responses to hard and very hard problems into length quartiles and plot accuracy (Figure 4b).

Accuracy falls steadily with length even under fixed difficulty. On hard problems, most models peak in the mid-length range (12k–16k tokens, Table 5), suggesting a sweet spot: long enough for reasoning but short enough to avoid over-thinking. Beyond this, accuracy drops consistently. GFPO variants outperform GRPO in the longest bin (67% vs 52% on Hard; 20% vs 17% on Very Hard, Table 6), as their longest responses are both shorter (20k vs 24k on Hard; 27k vs 28k on Very Hard) and more accurate. On very hard problems, degradation is sharper. Adaptive Difficulty and Token Efficiency briefly improve from short to mid-low bins, but all methods decline at longer lengths. Token Efficiency and Shortest 8/24 show the steepest drops, likely from reduced exposure to long chains. Adaptive Difficulty is the most robust, maintaining stable accuracy across bins. By contrast, SFT degrades little with length but rarely solves hard problems, producing a flat yet low curve.

We complement this with an absolute-length analysis across models (Figure 4a). GFPO shifts substantial mass away from the long tail ($\geq$20k tokens), cutting it from 32% under GRPO to 22–23%, and increasing the share of <15k responses. These shorter chains often solve harder problems: in the $\leq$5k bin, GFPO's prompt difficulty is $\sim$9$\times$ higher than GRPO's (16–18% vs 2%) with only minor accuracy loss (100% $\rightarrow$ 97%). Lower accuracy in GFPO's longest bins reflects that most solvable prompts are handled in shorter bins; the remaining long chains correspond to the hardest cases.

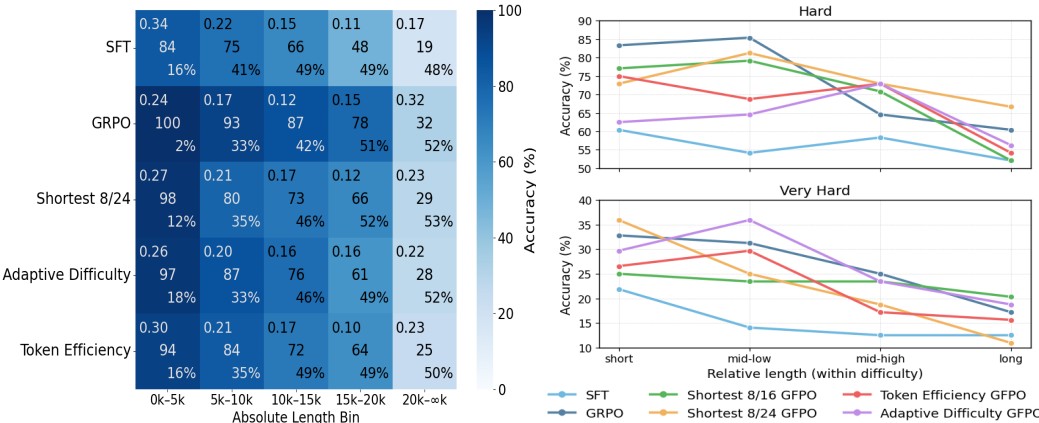

(a) **Accuracy, Response Share, and Prompt Difficulty by Response Length.** Each cell shows accuracy (center), response share (top left), and prompt difficulty (bottom right; avg difficulty $(1 - SFT_{acc})$ of prompts corresponding to responses in cell, for a fixed response length range.

(b) **Accuracy vs Relative Length for Hard and Very Hard Problems.** On very hard problems, Adaptive Difficulty is most robust. Token efficiency and Shortest 8/24 drop in the longer bins, likely due to aggressive filtering.

Figure 4: **Accuracy Across Response Lengths for AIME 25. (a)** GFPO cuts long-tail verbosity (32% to 22% outputs $\geq$ 20k tokens) and solves hard problems with shorter responses ($\sim$9x harder prompts solved with $\leq$ 5k tokens). **(b)** Accuracy declines with increasing response length even at fixed difficulty. On hard problems, most models peak at 12k-16k tokens, while GFPO variants outperform GRPO in the longest bin by producing shorter, more accurate long responses.

| Method | Step Time (m) | % ↑ Step Time | Latency (s) | % ↓ Latency | % ↓ Latency Overhead |
|---|---|---|---|---|---|
| SFT | — | — | 196.8 | — | — |
| GRPO | 28.5 | 0.0% | 315.1 | 0.0% | — |
| 8 of 16 | 35.8 | 25.7% | 272.9 | 13.4% | 35.7% |
| Token Eff. | **30.4** | **6.8%** | **225.0** | **28.6%** | **76.2%** |
| Adaptive Diff. | 36.8 | 29.5% | 255.7 | 18.9% | 50.2% |

Table 3: **Train–Test Trade-off.** Training step time vs. end-to-end latency for GRPO and GFPO variants. Token Efficiency GFPO reduces latency by $\sim$29% with only a 7% increase in training time, eliminating three-quarters of the latency overhead introduced by GRPO over SFT.

Together, the relative and absolute-length analyses show verbosity—not difficulty—is the main driver of GRPO's long-chain errors. GFPO mitigates this by solving harder problems more succinctly while maintaining or improving accuracy. Among variants, Shortest 8/24 and Adaptive Difficulty achieve the best balance—substantially shortening responses while preserving performance. Further gains may be possible by tuning the $k/G$ ratio for Token Efficiency and Adaptive Difficulty.

### 5.3 Train vs Test-time Trade-off.

We compare GRPO and several GFPO variants on training cost and inference latency (Table 3; see Appendix A.4 for details on calculations). Latencies are averaged over AIME 24, AIME 25, and GPQA, capturing the long tail of hard problems. GRPO slows inference dramatically, raising latency from 196.8 seconds for SFT to 315.1 seconds. Although GFPO samples 2x the responses of GRPO, we find this cost is almost completely offset as the learned policy produces shorter responses than GRPO at most time steps. Token Efficiency GFPO delivers nearly identical training cost (+6.8% step time, $\sim$3.2 extra hours) while cutting latency by 28.6% (315.1 s $\rightarrow$ 225.0 s), eliminating over three-quarters of GRPO's overhead relative to SFT. In practical terms, users wait about 90 seconds less per hard query—a substantial improvement. Other GFPO variants also reduce latency but at 29–66% higher training cost. Token Efficiency GFPO provides the clearest Pareto improvement: much faster responses for nearly the same training time.

**5.4  What is GFPO trimming?**  To analyze where GFPO's length savings arise, we annotate AIME 25 traces from five models—SFT, GRPO, Shortest 8/24 GFPO, Token Efficiency GFPO, and Adaptive Difficulty GFPO—using GPT-4o. Each trace is segmented into *Problem* (problem setup), *Solution* (developing candidate solutions), *Verification* (checking intermediate results), and *Final* (answer statements). GRPO inflates mid-trace reasoning compared to SFT—on AIME 25 the Solution segment expands from 6.5k to 8.3k tokens, and Verification from 1.9k to 3.1k (Figure 5). GFPO reverses this trend: Shortest 8/24 GFPO shrinks the Solution phase from 8.3k to 6.6k tokens ($\downarrow$ 94.4% of excess length), trimming many digressive solution attempts. It also reduces Verification from 3.1k to 2.3k tokens ($\downarrow$ 66.7% of excess length), cutting the redundant, circular checks in GRPO.

## 6  RELATED WORK

**GRPO Loss Modifications.** Several works refine GRPO's loss normalization to better handle token efficiency and stability. Dr. GRPO (Liu et al., 2025) normalizes by the longest chain in the batch, and DAPO (Yu et al., 2025) by the total token count—both amplifying penalties on long incorrect outputs. GFPO adopts DAPO's normalization (as in `verl` (Sheng et al., 2024), `TRL` (von Werra et al., 2020)) but shows that normalization alone cannot prevent verbosity: it penalizes long failures yet also rewards long successes. GFPO instead modifies the advantage function—filtering which chains count for learning—orthogonal to normalization and compatible with variants like Dr. GRPO.

**Length-Aware Penalties.** Another line of work directly penalizes verbosity, e.g., capping rewards beyond a token limit (Hou et al., 2025), applying adaptive or solve-rate scaled penalties (Su & Cardie, 2025; Xiang et al., 2025), or optimizing toward target lengths (Aggarwal & Welleck, 2025). Such reward engineering can reduce length but often harms accuracy or requires careful tuning. GFPO sidesteps explicit penalties: its rejection step implicitly shapes which outputs drive learning.

**Inference-Time Interventions.**  Reasoning length can also be managed without retraining. Prior work includes voting over the shortest $m$ of $k$ responses (Hassid et al., 2025), using "budget forcing" phrases to stop generation (Muennighoff et al., 2025), or halting once answers stabilize (Liu & Wang, 2025; Yang et al., 2025). These inference-time approaches are complementary to GFPO.

**Rejection Sampling Methods.** Rejection sampling has been applied post-training for length control, e.g., fine-tuning or DPO on shortest correct outputs (Kim et al., 2024), or stabilizing RL updates via prompt filtering (Yu et al., 2025; Xiong et al., 2025). GFPO integrates rejection

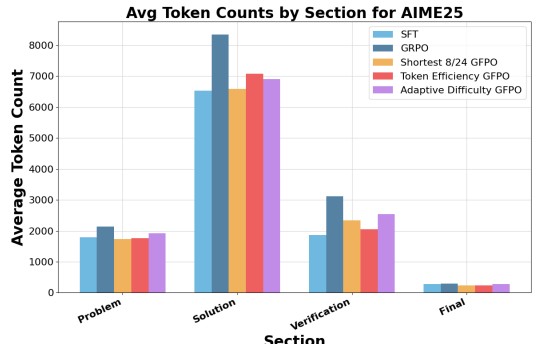

Figure 5: **Average Token Counts by Reasoning Segment.** GRPO inflates the Solution and Verification phases relative to SFT. GFPO variants markedly reduce this excess. On AIME 25, Shortest 8/24 cuts Solution length inflation by 94.4% and Verification by 66.7%.

within RL itself: sampling larger groups and updating only on top-ranked chains by length or reward efficiency. Related methods use rejection for contrastive pair selection (Khaki et al., 2024), constrained decoding (Lipkin et al., 2025), or efficient Best-of-$N$ search (Sun et al., 2024). GFPO differs in baking this principle directly into the policy update, avoiding distillation or explicit penalties.

## 7  CONCLUSION

We present Group-Filtered Policy Optimization (GFPO), a simple extension of GRPO that samples more responses and trains only on those aligned with target attributes. This filtering serves as implicit reward shaping, curbing length inflation without complex reward design. Across multiple models and challenging benchmarks, GFPO preserves GRPO-level accuracy while improving efficiency: with only a 7% increase in training time, it lowers end-to-end inference latency by nearly 30%, yielding 90-second faster response times on hard STEM queries. GFPO shifts modest additional compute to training to deliver decisively faster test-time reasoning without sacrificing response accuracy.

## 8 REPRODUCIBILITY STATEMENT

We detail our training and evaluation setup in Section 3, including our model, baselines, training configurations and infrastructure, evaluation metrics and datasets. We provide further details on how we compute our training time and inference latency estimates in Appendix A.4.

## 9 ACKNOWLEDGMENTS

We would like to thank Guoqing Zheng, Suriya Gunasekar, Piero Kauffmann, Mojan Javaheripi, Safoora Yousefi, and the AI Frontiers team at Microsoft Research for valuable discussions and feedback throughout this project.

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

## A APPENDIX

### A.1 LLM USAGE.

We use LLMs for some editing and rephrasing of paper text.

### A.2 QUALITATIVE EXAMPLES

---

**AIME 25 II Problem 1: GRPO v/s Token Efficiency GFPO**

**User:**

Six points $A, B, C, D, E$ and $F$ lie in a straight line in that order. Suppose that $G$ is a point not on the line and that $AC = 26$, $BD = 22$, $CE = 31$, $DF = 33$, $AF = 73$, $CG = 40$, and $DG = 30$. Find the area of $\triangle BGE$.

**GRPO**

`<think>`... So now triangle $BGE$: $B = (18, 0)$, $G = (58, 24)$, $E = (57, 0)$. Its base $BE =$ distance from 18 to 57 = 39. But then what is the height? The triangle's base is along x-axis and $G$ has y coordinate 24, but note that $G$ is not vertically above the segment? Actually, base $BE$: endpoints $(18, 0)$ and $(57, 0)$. The distance from $G$ to line $BE$ is simply the absolute y coordinate if $G$'s x coordinate lies between 18 and 57? But here $x_G = 58$, so $G$ is to the right of $E$. But the formula for area using base and height with base along line is the distance from $G$ to the line containing $BE$. But the line containing $BE$ is horizontal ($y = 0$). So the distance is $|24| = 24$, regardless of the x-coordinate. So area $= \frac{1}{2} \cdot$ base $\cdot$ height $= \frac{1}{2} \cdot 39 \cdot 24 = 468$.

...

Wait, $39 \cdot 24 = 936$, half is 468. But wait, is that the answer? Let's check: base $BE$ length $= 57 - 18 = 39$, height $= 24$. So area $= \frac{39 \cdot 24}{2} = 468$.

...

But the problem's answer is expected to be an integer? Possibly yes. But let's check step 1: "Six points $A, B, C, D, E, F$ lie in a straight line in that order." Our computed positions are consistent: $A = 0$, $B = 18$, $C = 26$, $D = 40$, $E = 57$, $F = 73$. Check distances: $AC = 26$, $BD = 40 - 18 = 22$, $CE = 57 - 26 = 31$, $DF = 73 - 40 = 33$, $AF = 73$, correct. Now then triangles: $CG =$ distance from $C = (26, 0)$ to $G = (58, 24)$ equals $\sqrt{(32)^2 + 24^2} = \sqrt{1024 + 576} = \sqrt{1600} = 40$, and $DG = \sqrt{(58 - 40)^2 + 24^2} = \sqrt{18^2 + 576} = \sqrt{324 + 576} = \sqrt{900} = 30$, so correct. Now triangle $BGE$: vertices $B = (18, 0)$, $G = (58, 24)$, $E = (57, 0)$. The base $BE$ has length $57 - 18 = 39$, and the height is the vertical distance from $G$ to the x-axis is 24, so area$= \frac{39 \cdot 24}{2} = 468$. So answer is 468.

...

However, double-check: It might be that $G$ could have negative $y$. But then area is same magnitude, absolute value. Therefore answer: 468.

...

But wait, check problem statement: "Find the area of triangle $BGE$." Not necessarily with base $BE$ horizontal? But we computed that. I want to check if there's any trick: Possibly the triangle is not oriented with base along the line containing $B$ and $E$ because then the altitude would be the distance from $G$ to that line. And indeed the line $BE$ is horizontal, so the altitude is the absolute value of $G$'s $y$-coordinate. We computed $G$'s $y$-coordinate as 24. So final answer: 468.

...

Wait, but what about possibility $x = 58$? Let's double-check the $x$-coordinate from difference of circle equations: We had circles: $(x - 26)^2 + y^2 = 40^2 = 1600$, $(x - 40)^2 + y^2 = 30^2 = 900$. Subtracting: $(x - 26)^2 - (x - 40)^2 = 700$. Expand: $(x^2 - 52x + 676) - (x^2 - 80x + 1600) = -52x + 676 + 80x - 1600 = 28x - 924 = 700$, so $28x = 1624$, $x = 58$. So that's consistent. So yes. Thus final answer: 468. ...... `</think>`

**Token Efficiency GFPO**

`<think>`... Now we want the area of triangle $BGE$. The vertices are $B = (18, 0)$, $G = (58, 24)$ or $(58, -24)$, and $E = (57, 0)$. The area is given by $\frac{1}{2}$ absolute value of cross product of $\overrightarrow{BG}$ and $\overrightarrow{BE}$ etc. Let's compute vectors: $\overrightarrow{BG} = G - B = (58 - 18, 24 - 0) = (40, 24)$. $\overrightarrow{BE} = E - B = (57 - 18, 0 - 0) = (39, 0)$. Area $= \frac{1}{2}|\det(\overrightarrow{BG}, \overrightarrow{BE})| = \frac{1}{2}|(40 \cdot 0 - 24 \cdot 39)| = \frac{1}{2}|-936| = 468$. Thus answer: 468. So final answer: 468. ...... `</think>`

---

> **Comparison:**
>
> The GRPO response recomputes the area of $\triangle BGE$ six times, while Token Efficiency GFPO computes it only once.

---

### AIME 25 I Problem 8: GRPO v/s Token Efficiency GFPO

**User:**

Let $k$ be real numbers such that the system $|25 + 20i - z| = 5$ and $|z - 4 - k| = |z - 3i - k|$ has exactly one complex solution $z$. The sum of all possible values of $k$ can be written as $\frac{m}{n}$, where $m$ and $n$ are relatively prime positive integers. Find $m + n$. Here $i = \sqrt{-1}$.

**GRPO**

`<think>`... Equate squares: $(x - (4+k))^2 + y^2 = (x-k)^2 + (y-3)^2$. Expand: $(x - (4+k))^2 = (x - k - 4)^2 = (x-k)^2 - 2 \cdot 4 \cdot (x-k) + 16$, wait check: actually, let's expand properly. Let $A = 4 + k$, $B = k$, then equation becomes $(x - A)^2 + y^2 = (x - B)^2 + (y - 3)^2$. Expand LHS: $(x - A)^2 = x^2 - 2Ax + A^2$ and then plus $y^2$, so LHS $= x^2 - 2Ax + A^2 + y^2$. RHS: $(x-B)^2 + (y-3)^2 = (x^2 - 2Bx + B^2) + (y^2 - 6y + 9) = x^2 - 2Bx + B^2 + y^2 - 6y + 9$. Cancel $x^2$ and $y^2$ yields $-2Ax + A^2 = -2Bx + B^2 - 6y + 9$. Rearranging: $-2Ax + A^2 + 2Bx - B^2 + 6y - 9 = 0$, or $2(B - A)x + 6y + (A^2 - B^2 - 9) = 0$. But note that $A^2 - B^2 = (A + B)(A - B)$ and $A + B = (4 + k) + k = 4 + 2k$, and $A - B = (4 + k) - k = 4$, so $A^2 - B^2 = 4(4 + 2k) = 16 + 8k$. Then our equation is $2(B - A)x + 6y + (16 + 8k - 9) = 0$, i.e. $2(B - A)x + 6y + (7 + 8k) = 0$, but note that $B - A = k - (4 + k) = -4$. Then $2(B - A)x = 2(-4)x = -8x$, so the equation becomes $-8x + 6y + 7 + 8k = 0$ or $8x - 6y = 7 + 8k$.

...

Alternatively, we can get the perpendicular bisector of the segment connecting $(4 + k, 0)$ and $(k, 3)$. Let's check the midpoint: $M = \left(\frac{4+k+k}{2}, \frac{0+3}{2}\right) = \left(\frac{4+2k}{2}, \frac{3}{2}\right) = (2 + k, 3/2)$. The slope of the line joining these two points from $(4 + k, 0)$ to $(k, 3)$ equals $\frac{3-0}{k-(4+k)} = \frac{3}{-4} = -\frac{3}{4}$. So the perpendicular bisector slope is the positive reciprocal: $\frac{4}{3}$, and it passes through $M$. Its equation is $y - 3/2 = \frac{4}{3}(x - (2 + k))$. Multiply both sides by 3: $3y - 9/2 = 4(x - 2 - k)$. Multiply both sides by 2: $6y - 9 = 8(x - 2 - k) = 8x - 16 - 8k$. Then rearranging: $6y - 9 = 8x - 16 - 8k$, so $8x - 6y = 16 + 8k + 9 = 25 + 8k$.

...

Let's re-check... So the equation is $(x - (4 + k))^2 + (y - 0)^2 = (x - k)^2 + (y - 3)^2$. Expand LHS: $(x - 4 - k)^2 + y^2 = (x - k - 4)^2 + y^2 = (x^2 - 2x(4 + k) + (4 + k)^2) + y^2$. The RHS is $(x - k)^2 + (y - 3)^2 = x^2 - 2kx + k^2 + y^2 - 6y + 9$. Cancel $x^2$ and $y^2$ from both sides to get $-2(4 + k)x + (4 + k)^2 = -2kx + k^2 - 6y + 9$. Bring terms to one side: $-2(4 + k)x + (4 + k)^2 + 2kx - k^2 + 6y - 9 = 0$. Combine the $x$ terms: $-2(4 + k)x + 2kx = -8x - 2kx + 2kx = -8x$. Then the constant terms: $(4 + k)^2 - k^2 - 9$; expanding $(4 + k)^2 = k^2 + 8k + 16$, subtracting $k^2$ gives $8k + 16$, and subtracting 9 gives $8k + 7$. So the equation becomes $-8x + 6y + (8k + 7) = 0$, or equivalently $8x - 6y = 8k + 7$.
...... `</think>`

**Token Efficiency GFPO**

`<think>`... Squaring both sides yields: $(x - (4 + k))^2 + y^2 = (x - k)^2 + (y - 3)^2$. Expand: $(x^2 - 2(4 + k)x + (4 + k)^2 + y^2) = (x^2 - 2kx + k^2 + y^2 - 6y + 9)$. Cancel $x^2$ and $y^2$, leaving: $-2(4 + k)x + (4 + k)^2 = -2kx + k^2 - 6y + 9$. Simplify: Bring terms: $-2(4 + k)x + (4 + k)^2 + 2kx - k^2 + 6y - 9 = 0$. Combine $x$ terms: $-2(4 + k)x + 2kx = -2(4 + k - k)x = -2 \cdot 4x = -8x$. So equation becomes: $-8x + (4 + k)^2 - k^2 + 6y - 9 = 0$. Simplify constant: $(4+k)^2 - k^2 = (16 + 8k + k^2) - k^2 = 16 + 8k$. So the equation is: $-8x + (16 + 8k) + 6y - 9 = 0$, or $-8x + 6y + (8k + 7) = 0$, but check: $16 + 8k - 9 = 7 + 8k$. Multiply by $-1$: $8x - 6y - (8k + 7) = 0$, or in standard form: $8x - 6y = 8k + 7$. ...... `</think>`

> **Comparison:**
>
> GRPO response recomputes and checks the perpendicular bisector equation 5 times (only 3 instances shown), while Token Efficiency GFPO computes it only once.

---

**AIME 25 II Problem 5: GRPO v/s Shortest 8/24**

**User:**

Suppose $\triangle ABC$ has angles $\angle BAC = 84°$, $\angle ABC = 60°$, and $\angle ACB = 36°$. Let $D$, $E$, and $F$ be the midpoints of sides $\overline{BC}$, $\overline{AC}$, and $\overline{AB}$, respectively. The circumcircle of $\triangle DEF$ intersects $\overline{BD}$, $\overline{AE}$, and $\overline{AF}$ at points $G$, $H$, and $J$, respectively. The points $G$, $D$, $E$, $H$, $J$, and $F$ divide the circumcircle of $\triangle DEF$ into six minor arcs, as shown. Find $\widehat{DE} + 2 \cdot \widehat{HJ} + 3 \cdot \widehat{FG}$, where the arcs are measured in degrees.

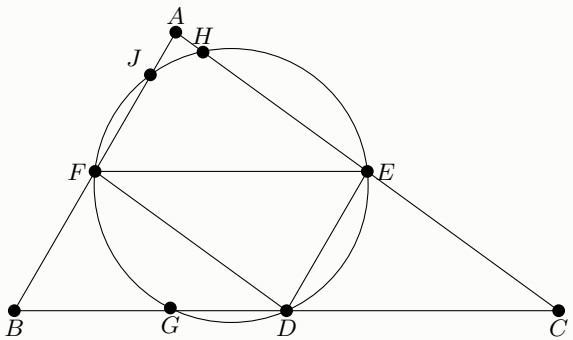

**GRPO**

`<think>` ...... We need to compute arcs on the circumcircle of triangle $DEF$. Let's denote the vertices of triangle $DEF$ in order: It says circle of triangle $DEF$, and then the points $G, D, E, H, J, F$ are arranged in order along circle. So the vertices of triangle $DEF$ are $D, E, F$. But note the circle has six points in order, and the six arcs are (arc $DE$), then arc $EH$? Wait, but check the given: They mention "the points $G, D, E, H, J,$ and $F$ divide the circumcircle of triangle $DEF$ into six minor arcs, as shown." That means the circle order is: $G$, then $D$, then $E$, then $H$, then $J$, then $F$, and then back to $G$. But we need to be careful: The circumcircle of triangle $DEF$ has vertices $D, E, F$ as well. So the arcs are between adjacent points in the circle order. The six arcs are: arc $GD$, arc $DE$, arc $EH$, arc $HJ$, arc $JF$, arc $FG$. But the problem statement says "the points $G, D, E, H, J,$ and $F$ divide the circumcircle of triangle $DEF$ into six minor arcs, as shown." But then they ask "find arc $DE + 2 \cdot$ arc $HJ + 3 \cdot$ arc $FG$." But wait, which arc is arc $FG$? The arcs, I need to check which arcs are which from the order: They are given in the diagram: The points in order are: $G$, then $D$, then $E$, then $H$, then $J$, then $F$, then back to $G$. So the arcs are: arc $GD$, arc $DE$, arc $EH$, arc $HJ$, arc $JF$, arc $FG$. But the problem asks for arc $DE$, then $2 \cdot$ (arc $HJ$), then $3 \cdot$ (arc $FG$). But note that arc $FG$ is the arc from $F$ to $G$, but in our ordering, the arc from $F$ to $G$ is the arc $FG$, but careful: the order is $G$-$D$-$E$-$H$-$J$-$F$, so arc $FG$ is the arc from $F$ to $G$, which is indeed the last arc in the sequence. So we need to compute these arc measures, but they are not trivial. We need to find angles related to triangle $ABC$ and midpoints $D, E, F$. Let coordinates: $A, B, C$. But maybe we can use properties: The circle through midpoints of sides of a triangle is the medial circle, but it's not necessarily the circumcircle of the medial triangle, but that's exactly what it is: The circumcircle of triangle $DEF$, where $D, E, F$ are midpoints of $BC, AC, AB$ respectively, is the nine-point circle of triangle $ABC$. ..... `</think>`

**Shortest 8/24 GFPO**

`<think>` ...... We need to compute arcs on the circumcircle of triangle $DEF$. Let's denote the vertices of triangle $DEF$ in order: It says circle of triangle $DEF$, and then the points $G, D, E, H, J, F$ are arranged in order along circle. So the vertices of triangle $DEF$ are $D, E, F$. But note the circle has intersections with lines $BD, AE, AF$. Let's re-read the problem statement. Given triangle $ABC$ with angles $A = 84$, $B = 60$, $C = 36$. $D, E, F$ are midpoints of sides $BC, AC, AB$ respectively. The circumcircle of triangle $DEF$ intersects $BD, AE, AF$ at points $G, H,$ and $J$ respectively. The points $G, D, E, H, J, F$ divide the circumcircle of triangle $DEF$ into six minor arcs, as shown. I want to label positions: We have triangle $ABC$, but then midpoints $D, E, F$ are on sides $BC, AC, AB$ respectively. Now, note that triangle $DEF$ is the medial triangle of $ABC$. And the circumcircle of the medial triangle is known to be the nine-point circle of triangle $ABC$. .... `</think>`

**Comparison:**

The Shortest 8/24 GFPO response quickly recognizes that $\triangle DEF$ is the medial triangle and its circumcircle is the nine-point circle, while the GRPO response gets bogged down in arc bookkeeping before arriving at the same conclusion.

---

### AIME 25 I Problem 9: GRPO v/s Adaptive Difficulty GFPO

**User:**

The parabola with equation $y = x^2 - 4$ is rotated 60 degrees counterclockwise around the origin. The unique point in the fourth quadrant where the original parabola and its image intersect has $y$-coordinate $\frac{a - \sqrt{b}}{c}$. Find $a + b + c$.

**GRPO**

```
<think> .....
```
But before that, we can attempt to factor the quartic if possible. It is a quartic in $u$. But perhaps a substitution might simplify it? Alternatively, note that $u$ might be a "nice" value? Alternatively, we can try to find the intersection by considering symmetry. Alternatively, consider the fact that the intersection point is the unique one in the fourth quadrant. Maybe we can try to solve directly by eliminating $u$. Alternatively, consider using rotation in reverse: For an intersection point $(x, y)$, we have $y = x^2 - 4$ and also $(y + 4)$??? Not sure.
...... `</think>`

**Adaptive Difficulty GFPO**

```
<think> .....
```
I think we need to solve for $x$ explicitly from the quartic. Alternatively, we can use the approach: Let $(x, y)$ be the intersection point. Then $(x, y)$ satisfies: $y = x^2 - 4$, and also $(x, y)$ satisfies the equation of the rotated parabola, i.e. if we rotate $(x, y)$ by $-60$ degrees, we get a point on the original parabola: So let $(u, v) = \left( \frac{1}{2}x + \frac{\sqrt{3}}{2}y, \ -\frac{\sqrt{3}}{2}x + \frac{1}{2}y \right)$. Then $y = u^2 - 4$.
...... `</think>`

**Comparison:**

The GRPO response jumps between several different alternatives—factoring the quartic, trying substitutions, symmetry arguments, and vague rotation ideas without committing to a plan and doesn't reduce uncertainty, whereas the Adaptive Difficulty GFPO defines a single clear approach to take.

---

**GPQA: GRPO v/s Token Efficiency GFPO**

**User:**

Consider this annihilation process $p + \bar{p} \rightarrow 2A^+ + 2A^-$
The antiproton is slowly moving and $m_A c^2 = 300 MeV$.
What is the velocity of particle A?

**GRPO**

`<think>`... Actually, maybe the process is in the lab frame where the antiproton is slow relative to the proton. But wait, which is slowly moving: "The antiproton is slowly moving" means that maybe the proton is moving? Wait check: "$p + \bar{p}$" one of them is slow, but which one? "The antiproton is slowly moving" means that the proton is moving fast? But then the question states: "Consider this annihilation process: $p + \bar{p} \rightarrow 2A^+ + 2A^-$". The wording "The antiproton is slowly moving" implies that the initial momentum is coming from the proton, which is fast. But then the question is asking: "What is the velocity of particle $A$?" Possibly meaning the outgoing particles have velocity determined by energy-momentum conservation in the center-of-mass frame.

...

Alternatively, the problem might be set in the center-of-mass frame. Let me think: "Annihilation process" usually is considered in the center-of-mass frame. But then "The antiproton is slowly moving" might be a hint that the proton is at rest? Because usually one would take $p$ is at rest, and the anti-proton is slow relative to $p$? Actually, "slowly moving" means small velocity. ... But if the antiproton is slow, then maybe the proton has kinetic energy? But then the process might be not exactly symmetric. Alternatively, maybe the assumption is that the antiproton is slow relative to the lab frame. That means the proton is at rest? Because then the antiproton is slow? That is contradictory: if the proton is at rest, then the anti-proton must be moving if we are in the lab frame?

...

Maybe the intended scenario is that the annihilation takes place in the lab frame where the antiproton is almost at rest, and the proton is moving with a velocity such that the center-of-mass energy is just above threshold. But then threshold for production of $4A$ is $4m_A = 1200$ MeV.

...

That doesn't work. Maybe the intended scenario is that both the proton and antiproton are nearly at rest in the center-of-mass frame...
...... `</think>`

**Token Efficiency GFPO**

`<think>`... Possibly the antiproton is slowly moving, so the reaction is in the lab frame, and one particle is at rest, but then the total kinetic energy available is the difference of the rest masses? But careful: if the antiproton is "slowly moving" that means it is non-relativistic? But then the proton might be at rest or antiproton at rest? Let me re-read: "The antiproton is slowly moving." So perhaps the proton is at rest? Actually, which one is at rest? The phrase "slowly moving" might indicate that the antiproton is nearly at rest relative to the proton? But then the total energy available is the mass difference, but then momentum conservation forces the final products to share momentum such that they have equal velocity magnitude. But which velocity? Let me analyze the reaction in the center-of-mass frame. ...... `</think>`

**Comparison:**

The GRPO response repeatedly rehashes whether the reaction is in the lab frame or center-of-mass frame, while Token Efficiency GFPO response correctly makes the assumption of center-of-mass frame once.

---

### GPQA: GRPO v/s Adaptive Difficulty GFPO

**User:**

A quantum mechanical particle of mass $m$ moves in two dimensions in the following potential, as a function of $(r, \theta)$: $V(r, \theta) = \frac{1}{2}kr^2 + \frac{3}{2}kr^2 \cos^2(\theta)$. Find the energy spectrum.

---

**GRPO**

`<think>`... Let $x = r\cos\theta$, $y = r\sin\theta$. Then $r^2 = x^2 + y^2$, $\cos^2\theta = \frac{x^2}{r^2}$ if $r \neq 0$. So potential becomes: $V(x, y) = \frac{1}{2}k(x^2 + y^2) + \frac{3}{2}k(x^2 + y^2)\left(\frac{x^2}{x^2+y^2}\right) = \frac{1}{2}k(x^2 + y^2) + \frac{3}{2}kx^2 = \frac{1}{2}kx^2 + \frac{1}{2}ky^2 + \frac{3}{2}kx^2 = (2)kx^2$? Let's check: $\frac{1}{2}kx^2 + \frac{3}{2}kx^2 = 2kx^2$, plus $\frac{1}{2}ky^2$ remains. So potential becomes: $V(x, y) = 2kx^2 + \frac{1}{2}ky^2$. That is a two-dimensional harmonic oscillator with different frequencies along $x$ and $y$. But wait, does that match? Let's recalculate: Given $V(r, \theta) = \frac{1}{2}kr^2 + \frac{3}{2}kr^2\cos^2\theta = \frac{1}{2}k(x^2 + y^2) + \frac{3}{2}k\cos^2\theta(x^2 + y^2)$. But $\cos\theta = x/r$ so $\cos^2\theta = \frac{x^2}{x^2+y^2}$. So then $V = \frac{1}{2}k(x^2 + y^2) + \frac{3}{2}k(x^2 + y^2) \cdot \frac{x^2}{x^2+y^2} = \frac{1}{2}k(x^2 + y^2) + \frac{3}{2}kx^2$. So indeed $V = \frac{1}{2}ky^2 + \left(\frac{1}{2} + \frac{3}{2}\right)kx^2 = \frac{1}{2}ky^2 + 2kx^2$. That is a harmonic oscillator with frequencies: $\omega_x = \sqrt{\frac{4k}{m}} = 2\sqrt{\frac{k}{m}}$ and $\omega_y = \sqrt{\frac{k}{m}}$.
...

Wait, but our derivation assumed that the potential is separable in Cartesian coordinates, which is valid if we perform transformation $x = r\cos\theta$, $y = r\sin\theta$. But is that always valid? Let's double-check: The given potential is $V(r, \theta) = \frac{1}{2}kr^2 + \frac{3}{2}kr^2\cos^2\theta = \frac{1}{2}k(x^2 + y^2) + \frac{3}{2}k\left(\frac{x^2}{x^2+y^2}\right)(x^2 + y^2) = \frac{1}{2}k(x^2 + y^2) + \frac{3}{2}kx^2 = \frac{1}{2}ky^2 + 2kx^2$. Yes that is indeed a separable potential in Cartesian coordinates with $x$ and $y$ degrees of freedom. So the energy eigenvalues are those of two independent harmonic oscillators with frequencies $\omega_x = \sqrt{\frac{4k}{m}} = 2\sqrt{\frac{k}{m}}$ and $\omega_y = \sqrt{\frac{k}{m}}$.
...... `</think>`

---

**Adaptive Difficulty GFPO**

`<think>`... Let $x = r\cos\theta$, $y = r\sin\theta$. Then $\cos^2\theta = \frac{x^2}{r^2}$ if $r \neq 0$. So then potential becomes: $V(x, y) = \frac{1}{2}k(x^2 + y^2) + \frac{3}{2}k(x^2 + y^2)\left(\frac{x^2}{x^2+y^2}\right) = \frac{1}{2}k(x^2 + y^2) + \frac{3}{2}kx^2$. So rewriting: $V(x, y) = \frac{1}{2}ky^2 + \left(\frac{1}{2}k + \frac{3}{2}k\right)x^2 = \frac{1}{2}ky^2 + 2kx^2$. Wait, check: $x^2 + y^2$ multiplied by factor $\frac{1}{2}k$ gives $\frac{1}{2}kx^2 + \frac{1}{2}ky^2$ and then add $\frac{3}{2}kx^2$ gives total potential: $V(x, y) = \left(\frac{1}{2}k + \frac{3}{2}k\right)x^2 + \frac{1}{2}ky^2 = (2k)x^2 + \frac{1}{2}ky^2$. So then the potential is anisotropic harmonic oscillator in Cartesian coordinates with frequencies: for $x$: $\omega_x = \sqrt{\frac{4k}{m}} = 2\sqrt{\frac{k}{m}}$ and for $y$: $\omega_y = \sqrt{\frac{k}{m}}$.
...... `</think>`

---

**Comparison:**

The GRPO response recalculates the harmonic oscillator frequencies multiple times while the Adaptive Difficulty GFPO response only calculates this once.

---

**GPQA: GRPO v/s Shortest 8/24 GFPO**

**User:**

A light beam is propagating through a glass with index of refraction n. The glass is moving at constant velocity v in the same direction as the beam and toward the observer in laboratory. What is the speed of light in glass relative to the observer in laboratory? Take the speed of light in vacuum $c = 1$.

**GRPO**

`<think>`... Let me derive from special relativity: The speed of light in a medium is $c/n$ in the medium's rest frame. But when the medium is moving with velocity $v$, the speed of light in the lab frame is given by the relativistic velocity addition formula: $u' = \frac{u+v}{1+(uv)/c^2}$. Here, $u = c/n$ and $v$ is the velocity of the medium in lab frame. But wait, is it $v$? But careful: The medium is moving with velocity $v$ in lab frame and the light moves in the medium frame with speed $c/n$. But the medium's velocity relative to lab is $v$, so we can use velocity addition. But then the effective speed in lab frame for light traveling in the same direction as medium is: $u = \frac{c/n+v}{1+(vc/(nc^2))}$? Wait, let me recalc: The formula is: $u = \frac{u'+v}{1+u'v/c^2}$. But $c = 1$, so $u = \frac{1/n+v}{1+v/(n)}$. Actually, check: If $u' = 1/n$ then $u = \frac{1/n+v}{(n+v)/n} = \frac{1/n+v}{(n+v)/n} = \left(\frac{1}{n}+v\right)\frac{n}{n+v} = \frac{1+nv}{n+v}$. But wait, then answer is $\frac{1+nv}{n+v}$?
...
But is that the Fizeau drag effect? The Fizeau drag coefficient is $1 - 1/n^2$. Wait, I recall the formula for the speed of light in a moving medium in the direction of flow is: $u = 1/n + (1 - 1/n^2)v$. Let me check with our expression: $\frac{1/n+v}{1+v/n} = \frac{1/n+v}{(n+v)/n} = \left(\frac{1}{n}+v\right)\frac{n}{n+v} = \frac{1+nv}{n+v}$. Now expand that in $v$: Let $v$ be small. Write $\frac{1+nv}{n+v} = \frac{1+nv}{n(1+v/n)} = \frac{1}{n}\frac{1+nv}{1+v/n}$. Now expand denominator: $\frac{1}{1+v/n} = 1 - v/n + (v/n)^2 + \ldots$ So we get $= \frac{1}{n}(1+nv)(1-v/n) = \frac{1}{n}[1 + nv - v/n - nv \cdot (v/n)] = \frac{1}{n}[1 + nv - v/n - v^2]$. Actually, let me do it properly: $(1+nv)(1-v/n) = 1 - v/n + nv - v^2$. So that equals $1 + nv - v/n - v^2$. Dividing by $n$ gives $= 1/n + v - v/n^2 - v^2/n$. So the coefficient of $v$ is $v(1 - 1/n^2)$. So indeed, to first order in $v$, $u = 1/n + (1 - 1/n^2)v$. That is the Fizeau drag formula. So our expression is consistent with the known formula for the speed of light in a moving medium. But wait, are we using the correct sign? The medium is moving toward the observer, so the effective speed of light relative to the observer is increased relative to the medium's rest frame speed. So the answer is $u_{\text{lab}} = \frac{1/n+v}{1+v/n}$ which simplifies to $\frac{1+nv}{n+v}$.
...... `</think>`

**Shortest 8/24 GFPO**

`<think>`...
The transformation from glass frame to lab frame, where the speed of light is $c/n$ in the glass frame and then add the velocity of glass, which is $v$. But note that the direction is the same as the beam. But the lab observer sees the beam speed as $v + (1/n)$? But wait not exactly, because the addition formula is: $u' = \frac{u+v}{1+uv}$. But careful: Actually, we have an object moving at speed $u$ (with respect to the glass) and the glass moving at speed $v$ (with respect to lab) along same direction. The lab speed $u_{\text{lab}}$ of the light beam should be computed using relativistic addition formula: $u_{\text{lab}} = \frac{u_{\text{glass}}+v}{1+u_{\text{glass}}\cdot v}$. But note that $u_{\text{glass}}$ is $1/n$. So lab speed $= \frac{1/n+v}{1+(1/n)v}$. Multiply numerator and denominator by $n$: $= \frac{1+nv}{n+v}$ ...But in lab frame, the medium is moving, so the speed of light in lab frame is: $u = \frac{1/n+v}{1+v/n} = \frac{1+nv}{n+v}$ as computed. Wait check: Multiply numerator and denominator by $n$, get $\frac{1+nv}{n+v}$. Yes.
...... `</think>`

**Comparison:**

The GRPO response circles back and re-derives the velocity addition with $u_{\text{glass}} = 1/n$ multiple times, while the Shortest 8/24 GFPO response does this once with a small recheck.

## A.3 EXTENDED ACCURACY AND RESPONSE LENGTH ANALYSIS

| | AIME 25 | | AIME 24 | | GPQA | | Omni-MATH | | LiveCodeBench | |
|---|---|---|---|---|---|---|---|---|---|---|
| | Acc | Avg Len | Acc | Avg Len | Acc | Avg Len | Acc | Avg Len | Acc | Avg Len |
| SFT | 64.2 | 10.9k | 72.2 | 10.1k | 67.0 | 6.6k | 84.7 | 9.6k | 57.7 | 10.3k |
| GRPO | 72.4 | 14.8k | 77.7 | 13.3k | 67.5 | 10.7k | 86.0 | 12.7k | 56.7 | 13.9k |
| Dr. GRPO | 66.8 | 13.1k | 74.4 | 11.7k | 66.9 | 8.1k | 84.9 | 10.9k | 57.6 | 13.6k |
| 6 of 8 | 69.2 | 14.7k | 79.6 | 13k | 70.2 | 10.2k | 88.3 | 12.9k | 56.4 | 13.6k |
| 8 of 16 | **70.2** | 13.9k | **77.9** | 12.3k | **70.0** | 9.7k | **89.3** | 11.8k | **59.8** | 12.6k |
| 6 of 16 | 70.1 | 13.8k | 76.9 | 12.2k | 68.3 | 9.1k | 87.8 | 11.4k | 58.3 | **12.6k** |
| 4 of 16 | 69.7 | **13.3k** | 76.6 | **11.8k** | 68.6 | **8.8k** | 88.0 | **11.3k** | 57.2 | 12.3k |
| 8 of 24 | **70.4** | **12.6k** | 75.1 | 11.6k | 68.9 | 8.6k | 87.5 | 11.1k | 56.5 | **11.8k** |
| 6 of 24 | 68.5 | 13.1k | 75.6 | 11.9k | **70.2** | 8.7k | **88.1** | 10.9k | 58.7 | 12.4k |
| 4 of 24 | 70.3 | 13k | **76.5** | **11.3k** | 68.1 | **8.3k** | 87.6 | **10.5k** | 59.2 | 11.8k |
| Token Efficiency | 69.5 | **12k** | 76.4 | **10.6k** | 68.5 | **7.5k** | 87.4 | **10.1k** | 57.0 | **11k** |
| Adaptive Difficulty | **70.8** | 12.8k | **76.6** | 11.6k | **70.8** | 9k | **88.9** | 11.6k | **57.2** | 12.1k |

Table 4: **Pass@1 Accuracy and Average Response Lengths on AIME 25, AIME 24, GPQA, Omni-MATH, and LiveCodeBench.** GFPO variants substantially reduce response lengths while matching GRPO accuracy. We find no statistically significant differences in GFPO's accuracy under the Wilcoxon signed-rank test for any dataset. Dr. GRPO yields lower accuracy (66.6% vs 69.5% on AIME 25, 74.4% vs 76.4% on AIME 24, 66.7% vs 68.5% on GPQA) and substantially longer responses than GFPO (43.6% vs 70.9% len reduction on AIME 25, 48.5% vs 84.6% on AIME 24, 65.1% vs 79.7% on GPQA, and 7.2% vs 79.7% on LiveCodeBench).

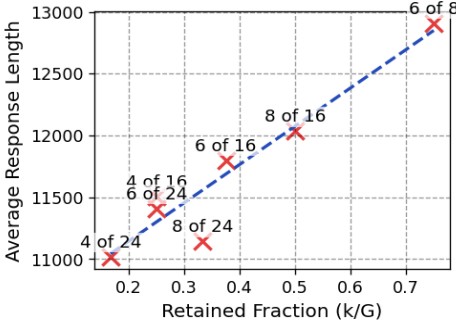

Figure 6: **Average Response Length vs k/G.** Reducing $k/G$, reduces average response length but beyond a point leads to diminishing returns.

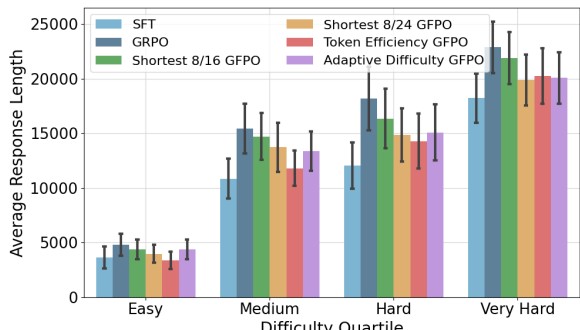

Figure 7: **Average Response Length Across Problem Difficulties.** Response lengths rise with problem difficulty for all methods, but GFPO reduces response length over all difficulty levels.

| Difficulty Bin | Method | Short | Mid-Low | Mid-High | Long |
|---|---|---|---|---|---|
| Hard | SFT | 7298 | 9949 | 12576 | 18349 |
| Hard | GRPO | 12719 | 16292 | 19846 | 23834 |
| Hard | Shortest 8/16 | 11292 | 13948 | 17897 | 22211 |
| Hard | Shortest 8/24 | 10087 | 12839 | 15711 | 20837 |
| Hard | Token Efficiency | **8918** | **12044** | **15337** | **20815** |
| Hard | Adaptive Difficulty | 9593 | 12959 | 16126 | 21677 |
| Very Hard | SFT | 10707 | 15630 | 20875 | 25666 |
| Very Hard | GRPO | 16728 | 22026 | 25309 | 27462 |
| Very Hard | Shortest 8/16 | 15768 | 20786 | 24051 | 26935 |
| Very Hard | Shortest 8/24 | **12657** | **18219** | 22671 | **25911** |
| Very Hard | Token Efficiency | 13034 | 18633 | 23223 | 26109 |
| Very Hard | Adaptive Difficulty | 13096 | 18625 | **22276** | 26279 |

Table 5: **Average Response Length by Difficulty and Length Bins for AIME 25.** We bin each model's responses to hard and very hard problems into length quartiles (short, mid-low, mid-high, long) and report the average response lengths across length bins. We **highlight** the shortest average response length per response length quartile across the different RL methods.

| Difficulty Bin | Method | Short | Mid-Low | Mid-High | Long |
|---|---|---|---|---|---|
| Hard | SFT | 60.42 | 54.17 | 58.33 | 52.08 |
| Hard | GRPO | **83.33** | **85.42** | 64.58 | 60.42 |
| Hard | Shortest 8/16 | 77.08 | 79.17 | 70.83 | 52.08 |
| Hard | Shortest 8/24 | 72.92 | 81.25 | **72.92** | **66.67** |
| Hard | Token Efficiency | 75.00 | 68.75 | **72.92** | 54.17 |
| Hard | Adaptive Difficulty | 62.50 | 64.58 | **72.92** | 56.25 |
| Very Hard | SFT | 21.88 | 14.06 | 12.50 | 12.50 |
| Very Hard | GRPO | 32.81 | 31.25 | **25.00** | 17.19 |
| Very Hard | Shortest 8/16 | 25.00 | 23.44 | 23.44 | **20.31** |
| Very Hard | Shortest 8/24 | **35.94** | 25.00 | 18.75 | 10.94 |
| Very Hard | Token Efficiency | 26.56 | 29.69 | 17.19 | 15.63 |
| Very Hard | Adaptive Difficulty | 29.69 | **35.94** | 23.44 | 18.75 |

Table 6: **Accuracy (%) by Difficulty and Length Bins for AIME 25.** We bin each model's responses to hard and very hard problems into length quartiles (short, mid-low, mid-high, long) and report the accuracies across length bins. We **highlight** the highest accuracy per response length quartile across the different RL methods.

## A.4 TRAIN TIME AND INFERENCE LATENCY ESTIMATION

**Training step times.** Each training run was executed for 100 steps. We report the *average wall-clock step time* by taking the mean over the 100 per-step times. This corresponds to end-to-end step duration including data loading, optimization, and logging. We train our models on 32xH100s, with a global batch size of 64, and 32,768 context. See §3 for more details.

**Inference latency.** Latencies are computed from evaluation runs on AIME 24, AIME 25, and GPQA. For AIME 24/25 we sample 32 responses per question and for GPQA we sample 5. For each response, the evaluation harness records a response_time equal to wall-clock time from request submission to receipt of the final token. We report average latency over all evaluation responses on AIME 24, 25, and GPQA. All evaluations were served with vLLM on 8×H100 GPUs with bfloat16 and maximum context length of 32,768. We used a single-GPU vLLM server per device with max_num_batched_tokens=65,536. All models were evaluated with identical decoding parameters (temperature, top-p, EOS/pad ids, and maximum length). See §3 for more details.

A.5   TUNING AND SCALING RETENTION RATIO $k/G$

Our results demonstrate that the retention ratio $k/G$ is a critical hyperparameter for tuning the length reductions derived by GFPO. For practitioners, our findings translate into simple guidelines for choosing $k$ and $G$ in new settings. We recommend starting with a retention fraction $k/G \approx 25 - 33\%$, which consistently delivers large length reductions with limited accuracy loss across math, coding, and STEM tasks. Practitioners may start with the largest group size $G$ that fits within their training budget, then choose $k$ so that $k/G$ lies in the 25–33% range (using a higher ratio for harder datasets to allow for more exploration). Finally, while $k/G$ remains a tunable hyperparameter, we find that strong rejection metrics such as token-efficiency filtering already provide substantial gains even at conventional ratios (e.g. $k/G = 50\%$), and in practice reduce sensitivity to the exact choice of $k/G$.

We explore scaling $G$ from $8 \to 16 \to 24$ and increasing $k$ up to $8$. As shown in Figure 6, average response length varies almost linearly with $k/G$ across these settings. We do not experiment with $k > 8$ because this would exceed the effective group size of our GRPO baseline (Phi-4-Reasoning-Plus, trained with $G = 8$), making comparisons unfair. Our findings indicate that the benefits of GFPO do not require scaling to very large values of $k$ or $G$:

- **Diminishing returns for large** $G$**.** Once $G$ is moderately large, further increases offer little additional gain. For example, $4/16$ and $6/24$ achieve nearly identical trace lengths.

- $k/G$ **is the control knob.** The same ratio can be obtained at multiple $G$ values (e.g., $4/16 \approx 6/24$), allowing practitioners to fix $G$ at a moderate value (e.g., 16) and tune $k$.

- **Better filtration beats bigger groups.** Stronger selection metrics outperform brute-force scaling. For instance, Token-Efficiency $8/16$ GFPO achieves larger length reductions than Shortest $8/24$, despite using a smaller group.

- **Moderate group sizes already provide stable advantages.** We agree that, in principle, larger $G$ and $k$ reduce variance in advantage estimation. However, our empirical results— and prior work—show that $G = 8$ or $16$ (Shao et al.) and even $G = 4$ (Qian et al.) already yield stable advantages and strong downstream performance.

A.6   ADAPTIVE DIFFICULTY ESTIMATES

**Algorithm 1** ADAPTIVE DIFFICULTY SAMPLING

---

**Require:** group $\mathcal{G} = \{o_1, \ldots, o_G\}$, t-digest tracker $\mathcal{T}$, reward function $R(\cdot)$, prompt $q$, scoring $metric(\cdot)$, sort $order$ ($\uparrow/\downarrow$)
1: $\mu_R \leftarrow \frac{1}{G} \sum_{i=1}^{G} R(q, o_i)$
2: $\mathcal{T}.\text{UPDATE}(\mu_R)$
3: **if** $\mathcal{T}.\text{READY}()$ **then**
4:     $(q_{25}, q_{50}, q_{75}) \leftarrow \mathcal{T}.\text{PERCENTILE}([25, 50, 75])$
5:     **if** $\mu_R < q_{25}$ **then** $k \leftarrow k_{\text{very-hard}}$
6:     **elif** $\mu_R < q_{50}$ **then** $k \leftarrow k_{\text{hard}}$
7:     **elif** $\mu_R < q_{75}$ **then** $k \leftarrow k_{\text{med}}$
8:     **else** $k \leftarrow k_{\text{easy}}$
9: **else**
10:     $k \leftarrow k_{\text{very-hard}}$
11: **end if**
12: $(\mathcal{S}, m) \leftarrow \text{REJECTIONSAMPLE}(\mathcal{G}, k, metric, order)$
13: **return** $\mathcal{S}, m$

---

Algorithm 1 details Adaptive GFPO. We tested EMA-style smoothing for per-question difficulty, but ultimately opted for a momentum-free approach. Given that difficulty evolves throughout training, meaningful adaptation requires a small momentum coefficient—making the method effectively equivalent to using the most recent estimate. Since our method bins questions into difficulty categories, only boundary-adjacent questions are sensitive to noise in their difficulty estimates. We thus use the simpler momentum-free variant, which adapts faster and avoids storing per-question state.

## A.7 TRAINING DYNAMICS

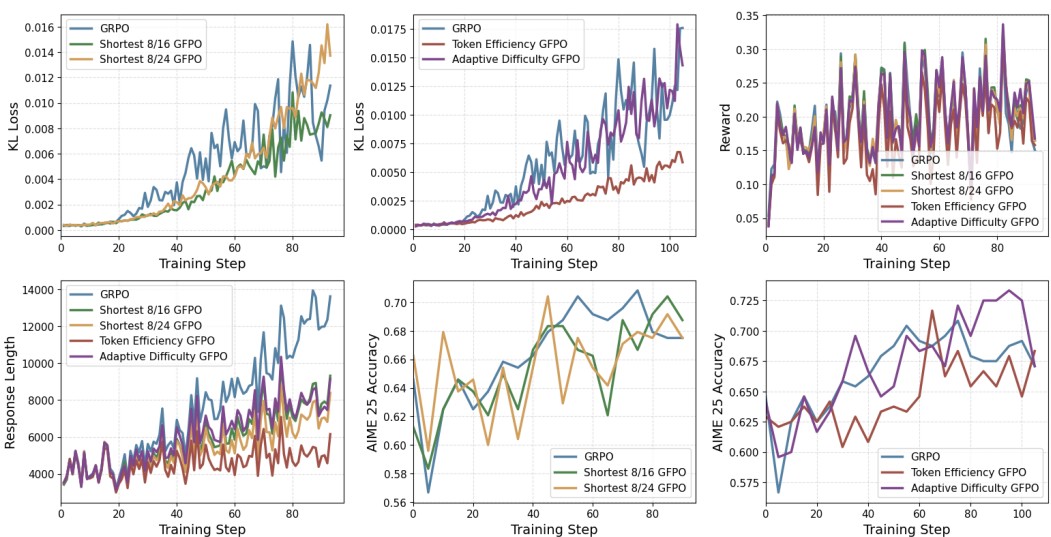

Figure 8: **GFPO vs GRPO Training Dynamics**. We observe no sudden KL explosions or collapses during GFPO training—in fact GFPO's KL curves are often as smooth or smoother than GRPO's. All GRPO and GFPO methods incur some reward oscillations over the course of training. But all methods monotonically converge towards the same reward, while GFPO substantially suppresses response length growth. We observe that GFPO tends to achieve its peak accuracy earlier than GRPO, while matching or improving on GRPO's best accuracy in almost all cases.

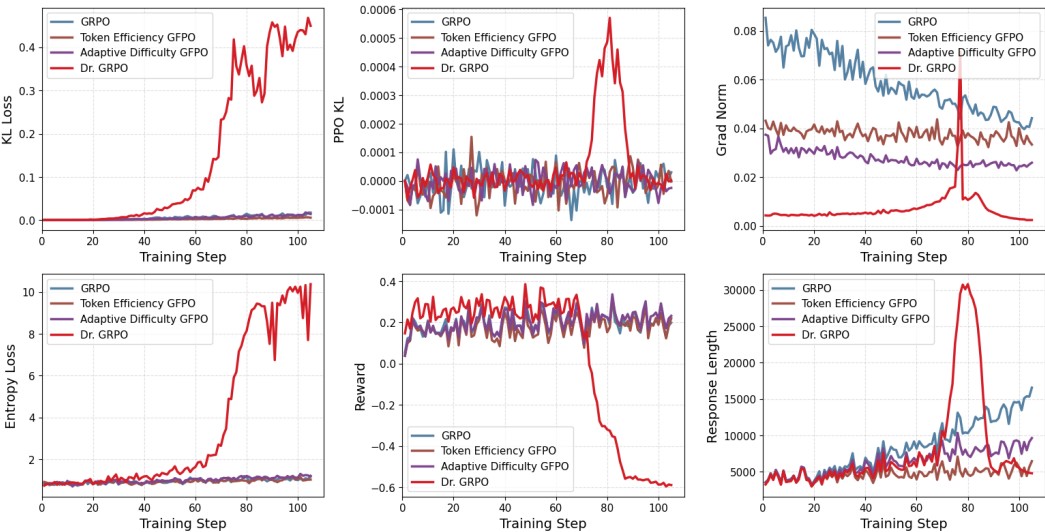

Figure 9: **GFPO vs Dr. GRPO Training Dynamics**. We observe occasional spikes in gradient norm, KL, and clip fraction when training with Dr. GRPO. These arise when a batch contains a few very long trajectories with large (unnormalized) advantages: because Dr. GRPO drops both std normalization and per-response length normalization, such trajectories dominate the update and cause transient but sharp jumps in the PPO statistics which can cause training to diverge. Meanwhile GFPO inherits the stable learning curves of GRPO.

