# OpenReview forum: "Sample More to Think Less: Group Filtered Policy Optimization for Concise Reasoning"
_ICLR.cc/2026/Conference — ICLR 2026 Poster_

### Official Review · Reviewer_g7JB · 2025-10-27

**Soundness:** 3
**Presentation:** 3
**Contribution:** 3
**Rating:** 6
**Confidence:** 4

**Summary:**

This paper introduces Group Filtered Policy Optimization, a simple modification to GRPO for RL with verifier rewards: for each question, sample a larger group of trajectories, rank by a target metric (e.g., length or reward-per-token), and update only on the top-k while zeroing advantages for the rest. Experimental results on Phi-4 reasoning demonstrate the effectiveness in reducing the generation length, latency, and training time.

**Strengths:**

1. A simple yet effective approach to dynamically adjusting the group size can broaden the response pool, incorporating more candidates with desirable traits and thereby optimizing toward the desired response properties.

2. The authors present comprehensive experimental results, evaluating not only mathematical tasks but also out-of-domain scenarios, and reporting both token-level efficiency and pass@1 accuracy.

3. The authors demonstrate that GFPO reduces latency by generating shorter responses than GRPO, while the token-efficiency variant achieves this with nearly the same training cost.

**Weaknesses:**

1. There is a line of research on token-efficient GRPO methods [1–4]. Although the authors mention several of these works in the related work section, they do not provide a direct comparison in the paper. Including such comparisons would strengthen the paper’s results and claims.

2. Additional training costs introduced during the process may limit the use of large values of G or K in GFPO.


[1] Aggarwal, Pranjal, and Sean Welleck. "L1: Controlling how long a reasoning model thinks with reinforcement learning." arXiv preprint arXiv:2503.04697 (2025).

[2] Luo, Haotian, et al. "O1-pruner: Length-harmonizing fine-tuning for o1-like reasoning pruning." arXiv preprint arXiv:2501.12570 (2025).

[3] Huang, Chengyu, Zhengxin Zhang, and Claire Cardie. "HAPO: Training Language Models to Reason Concisely via History-Aware Policy Optimization." arXiv preprint arXiv:2505.11225 (2025).

[4] Arora, Daman, and Andrea Zanette. "Training language models to reason efficiently." arXiv preprint arXiv:2502.04463 (2025).

**Questions:**

1. What is the performance of GFPO on the Qwen series model?

2. Could you conduct an ablation study by scaling up K (e.g., to 64 or 128)?

---

> ### Author Response · Authors · 2025-11-25
> **Official Comment by Authors**
>
> We thank reviewer g7JB for their thoughtful review and feedback. We appreciate that they find GFPO to be simple yet effective and our results to be comprehensive and generalizable to out-of-domain scenarios.
>
> Reviewer g7JB asks about the performance of GFPO on Qwen models and suggests experimenting with larger values of G or k. We conduct additional experiments for GFPO on Qwen models and share our results below. We also provide a detailed discussion regarding scaling k and G.
>
> Please let us know if you have any other questions or feedback during the discussion period. Thank you!
>
> **"What is the performance of GFPO on the Qwen series model?"**
>
> To address this question, we evaluate GFPO on the DeepSeek-R1-Distill-Qwen-7B and DeepSeek-R1-Distill-Qwen-14B models to determine if GFPO generalizes to other model sizes (beyond 14B) and model families (beyond Phi). DeepSeek-R1-Distill-Qwen-7B is a state-of-the-art 7B model obtained by supervised fine-tuning of Qwen2.5-Math-7B on DeepSeek-R1 reasoning traces. DeepSeek-R1-Distill-Qwen-14B is a strong alternative to the Phi-4-reasoning-plus model derived by supervised fine-tuning of Qwen2.5-14B on DeepSeek-R1 samples.
>
> We train these models with both GRPO (group size=8) and 8/16 shortest-k GFPO, where we sample 16 responses per question but only train on the shortest 8.
>
> **Training setup:** For the 7B model, we train with verl on 16 H100s with global batch size of 64 for 100 steps. We use Adam with learning rate=1e-6, cosine warmup for 10 steps, sampling temperature of T=1.0, KL regularization ($\beta=0.001$), and entropy coefficient ($\gamma=0.001$). We train with 16k context and reserve 1k tokens for the prompt.
>
> For the 14B model, we match the training setup of GFPO on the Phi-4-reasoning model and train with 32 H100s, with learning rate=1e-7, and 32k context. Other hyperparameters match the 7B setting.
>
> **Evaluation Setup:** We compare the resulting RLed models with the base DeepSeek-R1-Distill-Qwen-7B and DeepSeek-R1-Distill-Qwen-14B models (denoted as SFT) across AIME 25 (with 32 samples), AIME 25 (with 32 samples), and GPQA (with 5 samples). Responses are sampled at T=0.6, top-p=0.95 as recommended for the Deepseek Distill Qwen Models, with 16k max token length for the 7B model and 32k max token length for the 14B model.
>
> As in our prior experiments, we report pass@1 accuracy, raw response length, and excess length reduction, which captures the fraction of length increase of GRPO eliminated by GFPO.
>
> ### DeepSeek-R1-Distill-Qwen-7B Results
> | Method   | AIME 25         |   |          |    AIME 24      |   |          | GPQA   |         |          |
> |----------|----------|----------|----------|----------|----------|----------|----------|----------|----------|
> |          | Acc      |Avg Rsp Len| % Len Inf ($\downarrow$)|Acc      |Avg Rsp Len| % Len Inf ($\downarrow$) |Acc      |Avg Rsp Len| % Len Inf ($\downarrow$) |
> | SFT  | 37.0 | 11.4k  | N/A | 49.5 | 10.7k | N/A| 47.4 | 7.5K | N/A |
> | GRPO  | 40.2  | 13.3k | 0 | 51.5 | 11.9k | 0 | 48.4 | 8.3k | 0 |
> | 8 of 16 GFPO | 39.1 | 11.9k | 63.0 | 52.4 | 11k | 39.6 | 48.0 | 7.9k | 48.9 |
>
> **For DeepSeek-R1-Distill-Qwen-7B, we find that GFPO reduces excess response length inflation by **63%**, **39.6%**, and **48.9%** on AIME 25, 24, and GPQA** while closely matching the accuracy of GRPO on each benchmark. This demonstrates that GFPO remains effective at curbing response length inflation while maintaining accuracy even on smaller models.
>
> ### DeepSeek-R1-Distill-Qwen-14B Results
> | Method   | AIME 25         |   |          |    AIME 24      |   |          | GPQA   |         |          |
> |----------|----------|----------|----------|----------|----------|----------|----------|----------|----------|
> |          | Acc      |Avg Rsp Len| % Len Inf ($\downarrow$)|Acc      |Avg Rsp Len| % Len Inf ($\downarrow$) |Acc      |Avg Rsp Len| % Len Inf ($\downarrow$) |
> | SFT  | 48.2 | 12.3k  | N/A | 64.0 | 10.8k | N/A| 56.4 | 7k | N/A |
> | GRPO  | 51.9  | 16.1k | 0 | 69.3 | 13.9k | 0 | 56.8 | 8.8k | 0 |
> | 8 of 16 GFPO | 52.0 | 13.7k | 61.9 | 68.5 | 12.5k | 44.9 | 57.4 | 8.4k | 22.8 |
>
> **For DeepSeek-R1-Distill-Qwen-14B, we observe that GFPO reduces excess response length by **61.9%**, **44.9%**, and **22.8%** for AIME 25, AIME 24, and GPQA** compared to GRPO, while matching GRPO's accuracy. This further demonstrates that GFPO is effective across model families.
>
> We have included these results in Section 4.7 of our work.

---

> ### Author Response · Authors · 2025-11-25
> **Official Comment by Authors**
>
> **"Additional training costs introduced during the process may limit the use of large values of G or K in GFPO...Could you conduct an ablation study by scaling up K (e.g., to 64 or 128)?"**
>
> We appreciate the reviewer's interest in understanding how GFPO behaves as G and k scale.
>
> To reiterate, **G** is the sampled group size and **k** is the retained size after filtration. Across all our experiments, the ratio k/G emerges as the primary factor governing response-length reduction—smaller ratios consistently yield shorter traces.
>
> **We already explore scaling G from 8 → 16 → 24 and increasing k up to 8** (Section 4.1, Table 1). As shown in Figure 6, Average Response Length varies almost linearly with k/G across these settings. We do not experiment with k > 8 because this would exceed the effective group size of our GRPO baseline (Phi-4-Reasoning-Plus, trained with G = 8), making comparisons unfair.
>
> While we understand the motivation to examine very large values of G and k (e.g., 64 or 128), **our findings indicate that the benefits of GFPO do not require such extreme scaling:**
> - **Diminishing returns for large G.** Once G is moderately large, further increases offer little additional gain. For example, 4/16 and 6/24 achieve nearly identical trace lengths.
>
> - **k/G—not G—is the true control knob.** The same ratio can be obtained at multiple G values (e.g., 4/16 ≈ 6/24), allowing practitioners to fix G at a moderate value (e.g., 16) and simply tune k.
>
> - **Better filtration beats bigger groups.** Stronger selection metrics outperform brute-force scaling. For instance, Token Efficiency 8/16 GFPO achieves larger length reductions than Shortest 8/24, despite using a smaller group.
>
> - **Moderate group sizes already provide stable advantages.** We agree that, in principle, larger G and k reduce variance in advantage estimation. However, our empirical results—and prior work—show that G = 8 or 16 (Abdin et al.; Shao et al.) and even G = 4 (Qian et al.) already yield stable advantages and strong downstream performance.
>
> **Given these observations, we do not expect qualitatively new behavior at G = 64 or 128—only substantially higher cost.**
>
> Unfortunately, these extreme settings are also beyond our compute budget. Table 2 shows that per-step time grows rapidly as we move from G = 8 → 16 → 24, and the trend is slightly super-linear. Even under a conservative linear extrapolation:
> - G = 64 → ~94 minutes/step
> - G = 128 → ~170 minutes/step
>
> **A 100-step run on 32 H100s at `$3/hr` per GPU would cost roughly `$15k` (G=64) and `$27k` (G=128), likely more in practice. These costs make such runs infeasible for us.**
> However this scaling burden is not unique to GFPO—GRPO would incur the same cost when increasing group size.
>
> **Since most RL-for-reasoning work today trains with G ≈ 8–16, pushing to G=64 or 128 would also have limited practical relevance for researchers with similar resource constraints.**
>
> Overall, we view it as a strength of GFPO that it achieves substantial length reductions without requiring extreme scaling—making the method both effective and practical for a wide range of compute budgets.
>
> We have included a discussion on tuning and scaling k/G in Appendix A.5. We hope that our experiments and discussion address reviewer g7JB's questions. We would be happy to answer any other questions or comments you may have. Thank you!
>
> **References:**
>
> Marah Abdin, Sahaj Agarwal, Ahmed Awadallah, Vidhisha Balachandran, Harkirat Behl, Lingjiao Chen, Gustavo de Rosa, Suriya Gunasekar, Mojan Javaheripi, Neel Joshi, Piero Kauffmann, Yash Lara, Caio César Teodoro Mendes, Arindam Mitra, Besmira Nushi, Dimitris Papailiopoulos, Olli Saarikivi, Shital Shah, Vaishnavi Shrivastava, Vibhav Vineet, Yue Wu, Safoora Yousefi, and Guoqing Zheng. Phi-4-reasoning technical report, 2025.
>
> Rulin Shao, Shuyue Stella Li, Rui Xin, Scott Geng, Yiping Wang, Sewoong Oh, Simon Shaolei Du, Nathan Lambert, Sewon Min, Ranjay Krishna, Yulia Tsvetkov, Hannaneh Hajishirzi, Pang Wei Koh, and Luke Zettlemoyer. Spurious Rewards: Rethinking Training Signals in RLVR, 2025.
>
> Cheng Qian, Emre Can Acikgoz, Qi He, Hongru Wang, Xiusi Chen, Dilek Hakkani-Tür, Gokhan Tur, and Heng Ji. ToolRL: Reward is All Tool Learning Needs, 2025.

---

> ### Comment · Reviewer_g7JB · 2025-11-25
>
> Thank you for your response. It would be beneficial to compare this work with the prior works I mentioned in the weaknesses section.

---

> > ### Author Response · Authors · 2025-11-26
> > **Official Comment by Authors**
> >
> > Thank you so much for your reply. We are finalizing our experimental results on Dr. GRPO---a prominent alternative approach to token efficient reasoning, and will include this as a baseline soon.

---

> ### Author Response · Authors · 2025-11-27
> **Official Comment by Authors**
>
> **"There is a line of research on token-efficient GRPO methods...they do not provide a direct comparison in the paper. Including such comparisons would strengthen the paper's results and claims."**
>
> We appreciate the reviewer's suggestion to include a direct comparison against other token-efficient GRPO methods. To address this question, **we compare GFPO with Dr. GRPO (Liu et al.) on the Phi-4-reasoning model**. Dr. GRPO targets token-efficient reasoning by eliminating the per-response length normalization in GRPO.
>
> **Training Setup.** We train Dr. GRPO with the same configuration as our GFPO and GRPO experiments on 14B models---with verl on 32 H100s with global batch size of 64 for 100 steps. We use Adam with learning rate=1e-7, cosine warmup for 10 steps, sampling temperature of T=1.0, KL regularization ($\beta=0.001$), and entropy coefficient ($\gamma=0.001$). We train with 32k context and reserve 1k tokens for the prompt.
>
> **Evaluation Setup.** We evaluate on AIME 25 (32 samples), AIME 24 (32 samples), GPQA (5 samples), and LiveCodeBench (LCB) (3 samples) with T=0.8, top-p=0.95 with 32k max token length and report pass@1 accuracy (Acc), raw response length (Avg Rsp Len), and excess length reduction (% Len Inf ($\downarrow$)).
>
> Below we summarize our results comparing GRPO, Dr. GRPO, and our best GFPO method (Tok-Eff GFPO). SFT indicates the base Phi-4-reasoning model (w/o any RL training). We use **bold** to indicate which of Dr. GRPO vs GFPO has the better accuracy, response length, and % excess length reduction across each dataset. For a comparison across all GFPO methods, please see Table 1 and Table 4.
>
> ### GFPO vs Dr. GRPO Results
>
> | Method   | AIME 25         |   |          |    AIME 24      |   |          | GPQA   |         |          |LCB| | |
> |----------|----------|----------|----------|----------|----------|----------|----------|----------|----------|-----|----------|----------|
> |          | Acc      |Avg Rsp Len| % Len Inf ($\downarrow$)|Acc      |Avg Rsp Len| % Len Inf ($\downarrow$) |Acc      |Avg Rsp Len| % Len Inf ($\downarrow$) |Acc|Avg Rsp Len| % Len Inf ($\downarrow$) |
> | SFT  | 64.2 | 10.9k  | N/A | 72.2 | 10.1k | N/A| 67.0 | 6.6k | N/A | 57.7 | 10.3k | N/A |
> | GRPO  | 72.4  | 14.7k | 0 | 77.7 | 13.3k | 0 | 67.5 | 10.7k | 0 | 57.0 | 12.1k | 49.4 |
> | Dr. GRPO | 66.8 | 13.1k | 43.6 | 74.4 | 11.7k | 48.5 | 66.9 | 8.1k | 65.1 | **57.6** | 13.6k | 7.2|
> | Tok-Eff GFPO | **69.5** | **12k** | **70.9** | **76.4** | **10.6k** | **84.6** | **68.5** | **7.5k** | **79.7** | 57.0 | **11k** | **79.7**|
>
> **Results.** We find that **Dr. GRPO yields lower accuracy** (66.6\% vs 69.5\% on AIME 25, 74.4\% vs 76.4\% on AIME 24, 66.7\% vs 68.5\% on GPQA) **and substantially longer responses than GFPO** (43.6\% vs 70.9\% excess len reduction on AIME 25, 48.5\% vs 84.6\% on AIME 24, 65.1\% vs 79.7\% on GPQA, and 7.2\% vs 79.7\% on LiveCodeBench).

---

> ### Author Response · Authors · 2025-11-27
> **Official Comment by Authors**
>
> Continuing discussion of GFPO vs Dr. GRPO from previous comment:
>
> **Intuitions.** This behavior stems from the way Dr. GRPO modifies GRPO. By removing GRPO’s per-response length normalization, **Dr. GRPO makes gradient magnitudes scale with the number of tokens in each trajectory.** **Long responses therefore dominate the update signal**—regardless of quality—while short, efficient responses have comparatively little influence. Because our models are already heavily SFTed on chain-of-thought data and naturally generate long, high-variance traces, **the policy primarily learns to distinguish between good and bad long trajectories rather than shifting probability mass toward shorter ones**, leading to the observed accuracy drop and relatively small decrease in response length.
>
> This dynamic may be regime-dependent. In cold-start RL, where early responses are short and shallow, amplifying longer trajectories may actually help reasoning emerge. But for mature reasoning models, Dr. GRPO strengthens exactly the behaviors we aim to curb. GFPO, by contrast, directly prioritizes concise, high-quality trajectories, making it better aligned with post-SFT RL training.
>
> **Training Dynamics.** **We find training with Dr. GRPO to be unstable—with occasional spikes in gradient norm, KL divergence, entropy, and response length.** These arise when a batch contains a few very long trajectories with large (unnormalized) advantages. Because Dr. GRPO drops both std normalization and per-response length normalization, such trajectories dominate the update and cause transient but sharp jumps in the PPO statistics which can cause training to diverge. Please see Figure 9 for training curves and more details.
>
> **Meanwhile GFPO inherits the stable learning curves of GRPO.** We observe no sudden KL explosions or collapses during GFPO training—in fact GFPO's KL curves are often as smooth or smoother than GRPO's. All GRPO and GFPO methods incur some reward oscillations over the course of training. But all methods monotonically converge towards the same reward, while GFPO substantially suppresses response length growth. We observe that GFPO tends to achieve its peak accuracy earlier than GRPO, while matching or improving on GRPO's best accuracy in almost all cases. Please see Figure 8 for training curves.
>
> We have added these results to Section 4.6 of the paper and including analysis on training stability in Appendix A.7.

---

> ### Author Response · Authors · 2025-11-27
> **Official Comment by Authors**
>
> Hi Reviewer g7JB,
>
> Thank you again for the time you’ve spent reviewing our work and for your helpful suggestions on how to improve it. We’ve added several new experiments and analyses to incorporate your feedback:
>
> - **Experiments with GFPO on DeepSeek-R1-Distill-Qwen-7B and DeepSeek-R1-Distill-Qwen-14B:** Table 2, Section 4.7
> - **Discussion on why GFPO works well even without scaling k and G:** Appendix A.5
> - **Comparison of GFPO vs Dr. GRPO baseline:** Section 4.6, Table 1, Table 4, Appendix A.7 Figures 8 and 9
>
> If you feel that we have adequately addressed your questions, we would be very grateful if you could consider updating your score. If you have any further questions or suggestions, we would be happy to discuss them. Thank you!

---

### Official Review · Reviewer_3LZ1 · 2025-10-30

**Soundness:** 3
**Presentation:** 2
**Contribution:** 3
**Rating:** 4
**Confidence:** 3

**Summary:**

This paper addresses the critical issue of length inflation in large language models trained with reinforcement learning, where models tend to produce excessively verbose outputs without corresponding accuracy gains. The authors propose Group Filtered Policy Optimization (GFPO), a simple yet powerful modification to the GRPO training process. The key idea is to sample a larger group of responses during training and selectively update the policy using only a filtered subset of the most desirable ones—based on metrics such as conciseness or token efficiency. This in-training filtering serves as an implicit form of reward shaping, effectively teaching the model to be more concise at inference time.

**Strengths:**

1.GFPO’s core idea of in-training filtering is clear and easy to implement. By sampling a larger pool of responses and selectively training on the best subset, it avoids the complexities of explicit reward engineering. Framing this as implicit reward shaping provides an intuitive and generalizable way to steer model behavior toward desirable attributes like conciseness.

2. The paper demonstrates a rare and highly desirable outcome: improving inference efficiency while maintaining or even enhancing accuracy. The reported gains—up to an 85% reduction in excess length and a ~30% reduction in end-to-end latency are substantial and consistent across challenging STEM and coding benchmarks.

3. The method generalizes well beyond the training domain. On the out-of-distribution LiveCodeBench benchmark, GFPO continues to reduce response length where GRPO inflates it, sometimes even improving accuracy.

**Weaknesses:**

1. The Adaptive Difficulty GFPO relies on the average reward of sampled responses to estimate problem difficulty. While conceptually clever, this heuristic can be unstable and noisy, especially early in training or on high-variance problems. Randomly poor rollouts may cause easy instances to be misclassified as “hard,” leading to inefficient resource allocation.

2. The method’s success hinges on the retention fraction k/G, which determines how aggressively responses are filtered. Although the paper explores different ratios empirically, it offers limited theoretical or heuristic guidance for setting this parameter in new settings. This may require costly tuning for practitioners.

3. GFPO’s core trade-off sample more to think less implies higher computational expense during training. While the authors show that a 7% increase in training time yields a 29% reduction in inference latency, this additional cost may still be significant in resource-constrained environments.

4. Filtering based on conciseness or shortness may inadvertently penalize useful but verbose reasoning paths. This could suppress exploration and hinder the model from discovering complex problem-solving strategies that are initially long but later refinable into concise solutions.

**Questions:**

1. How stable is the average-reward–based difficulty estimate throughout training? Did you explore smoothing or momentum-based approaches to mitigate early-stage noise?

---

> ### Author Response · Authors · 2025-11-24
> **Official Comment by Authors**
>
> We thank reviewer 3LZ1 for their review and constructive feedback. We appreciate that they find GFPO to be "easy to implement", "generalizing beyond the training domain", and demonstrating the "rare and highly desirable outcome" of improving inference efficiency while maintaining accuracy.
>
> Reviewer 3LZ1 had questions on estimating problem difficulties in Adaptive difficulty GFPO, heuristic guidance on tuning the k/G retention ratio parameter, GFPO's train vs test-time trade-off, and if GFPO may suppress useful verbosity in responses. We answer all of these questions below. Please let us know if you have any further questions or comments during the discussion period. Thank you!
>
> **"Adaptive Difficulty GFPO relies on the average reward of sampled responses to estimate problem difficulty...this heuristic can be unstable and noisy, especially early in training."**
>
> We appreciate the reviewer's concern. To clarify, in our approach the raw average reward of a question is never directly used to decide its compute budget. Instead we:
> - Compute an average reward for the question.
> - Insert that value into a t-digest structure. The t-digest is maintained over all observed average rewards and provides a quartile-accurate estimate of the distribution.
> - Then use the quartile of that value, not the value itself, to assign the question to one of four coarse difficulty bins (easy, medium, hard, very-hard).
>
> **This algorithm uses only the rank of the question in the global difficulty distribution, not its exact numerical difficulty, which naturally allows for two types of robustness to noisy rewards:**
> 1. **Global smoothing via t-digest.** The digest aggregates all past difficulty estimates into clustered centroids, producing highly stable estimates of the quartile boundaries once enough samples have been seen.
> 2. **Coarse discretization into difficulty bins.** Because each question only needs to be assigned to one of four large buckets, small noise in its difficulty estimate does not matter unless the question sits extremely close to a quartile boundary. Questions far from boundaries are completely insensitive to moderate noise.
>
> **To avoid early-training noise before the t-digest has sufficient mass, we include a brief warm-up phase where all questions use a fixed k=8** (mirroring our baseline GFPO setting of shortest 8/16). After this period, the t-digest boundaries are likely to be quite stable.
>
> **"How stable is the average-reward–based difficulty estimate throughout training? Did you explore smoothing or momentum-based approaches to mitigate early-stage noise?"**
>
> This is a great suggestion. We agree that temporal smoothing is a reasonable idea and did investigate momentum-based approaches in an early stage of experimentation. Specifically, we computed exponential moving averages for the difficulty estimates of each question and used these smoothed estimates in our t-digest:
>
> $d_{i,t} = (1 - \\alpha) d_{i,t-1} + \\alpha \\hat{d}_{i,t}$
>
> where $d_{i,t-1}$ is the previous difficulty estimate of question $i$ and $\\hat{d}_{i,t}$ is the difficulty estimate at the current step.
>
> These experiments revealed two important considerations:
>
> 1. **Difficulty is expected to evolve during training.**
> As the model improves, problems that were initially "hard" naturally become easier. To track such shifts,
> $\alpha$ must be relatively large (small momentum), otherwise the estimate lags behind the model's true problem solving ability. In practice, we found that the best-performing values of $\alpha$ were very close to 1—effectively relying most heavily on the most recent difficulty estimate.
>
> 2. **EMA added little benefit in practice.**
> Because our algorithm ultimately only uses the difficulty quartile of a question and not its absolute difficulty value, only a small subset of borderline questions falling close to the quartile boundaries can benefit from momentum averaging. Empirically we found that swiftly adapting difficulty estimates (low momentum) led to better performance than higher momentum values to reduce noise in the difficulty estimates of a few borderline questions.
>
> Given these observations, we opted for a **simpler momentum-free version of the Adaptive Difficulty algorithm** which cuts an unnecessary hyperparameter ($\alpha$), avoids maintaining a per-question difficulty table (which can be large depending on corpus size), and empirically performs equivalently.
>
> Finally, we find that **training under the current momentum-free version of Adaptive Difficulty is quite stable**. Appendix Figures 8 and 9 show KL loss, reward, response length, and accuracy curves for this method, all of which behave smoothly and without signs of instability.
>
> We have updated Section 2.1 to include more details on the Adaptive Difficulty GFPO algorithm (previously omitted for space) and incorporate the above discussion to clarify why momentum-free difficulty estimation is both simple and effective in Appendix A.6.

---

> ### Author Response · Authors · 2025-11-24
> **Official Comment by Authors**
>
> **"Although the paper explores different [k/G retention] ratios empirically, it offers limited theoretical or heuristic guidance for setting this parameter in new settings."**
>
> We appreciate the reviewer's question regarding guidance on selecting k/G. We provide actionable guidelines for practitioners and ground these in our experiment results:
>
> 1. **Practical rule-of-thumb: retain 25–33% of samples.**
> As shown in Figure 6 and Table 1, k/G values of 8/24 (33%), 4/16 (25%), 6/24 (25%), and 4/24 (16%) lead to the lowest response lengths. However, the additional excess len reduction in going from 33% to 16% retained samples is only ~4% suggesting that further reductions in k/G would yield diminishing returns. Thus our results suggest that retaining 25-33% of the responses can be a good rule of thumb. This range of k/G translates to substantial length reductions across a variety of benchmarks (math, coding, stem) suggesting that it can also be a good starting point for other new settings.
>
> 2. **For a fixed k/G ratio, larger group sizes are preferable.**
> When comparing configurations with the same ratio (e.g., 4/16 vs 6/24, both 25%), we find that the larger group size (6/24) yields slightly higher accuracy (72.2% vs 72.0%) and greater excess length reduction (47.1% vs 44.2%). This suggests a second practical guideline:
> Start with the largest group size G that fits within your  training budget, then choose k to place the k/G ratio in the 25–33% range (using a higher ratio for harder datasets to allow for more exploration -- similar to Adaptive GFPO).
>
> 3. **Strong filtering metrics reduce sensitivity to k/G.**
> Our experiments demonstrate that the token-efficiency rejection metric itself provides substantial length reductions even when the retention ratio is held fixed at a default value (e.g., 50% with 8/16). Token efficiency consistently outperforms shortest-k across a range of ratios, indicating that with a strong rejection metric, the exact value of k/G becomes less critical. This further makes GFPO easy to adopt: users can begin with a conventional ratio (e.g., 50%) and still obtain significant gains simply by switching to token-efficiency filtering.
>
> While we agree that k/G is ultimately a hyperparameter that may require tuning for optimal results (similar to learning rate), our empirical findings do provide clear and transferable heuristics: **(1) start with k/G $\\approx$ 25−33%, (2) use the largest group size G affordable, and (3) rely on token-efficiency filtering to reduce sensitivity to the exact k/G ratio.**
>
> We have added Appendix section A.5 to more explicitly outline our guidance on setting k/G to practitioners.

---

> ### Author Response · Authors · 2025-11-24
> **Official Comment by Authors**
>
> **"While the authors show that a 7% increase in training time yields a 29% reduction in inference latency, this additional cost may still be significant in resource-constrained environments."**
>
> We appreciate the reviewer’s concern regarding the 7% increase in training time. GFPO indeed follows a "sample more to think less" principle, and our results highlight a fundamental trade-off between training compute and inference-time efficiency.
>
> However, this trade-off is highly favorable—even in resource-constrained settings. In our setup (32xH100), training a 14B model with GRPO for 100 steps takes 47.5 hours. Meanwhile, training the same model with Token efficiency GFPO for 100 steps takes ~50.67 hours. **A 7% training time increase corresponds to a only ~3 additional hours of training. This cost is incurred once, whereas the 29% reduction in inference latency is realized every time the model is queried anywhere it is deployed.** Even conservatively, this yields substantial net savings when the model serves many users or handles repeated queries.
>
> Overall, we view GFPO’s compute trade-off as not only reasonable but beneficial, providing a small, one-time increase in training cost that enables large and persistent reductions in inference latency during real-world use.
>
> **"Filtering based on conciseness or shortness may inadvertently penalize useful but verbose reasoning paths."**
>
> We agree that many problems may genuinely benefit from extended reasoning. However, both prior work and our own analyses **consistently indicate that excessively long traces often correlate with worse—not better—reasoning quality.** Prior results highlight this pattern: Balachandran et al. (2025) report that DeepSeek-R1 produces solutions on AIME 25 that are 5 times longer than those of Claude 3.7 Sonnet without any corresponding improvement in accuracy.
>
> Our controlled analysis on AIME 25 using Phi-4-Reasoning-Plus (Abdin et al., 2025) further reinforces this trend: when comparing correct vs. incorrect attempts on the same AIME-25 question, the longer response is more likely to be wrong 72% of the time. This suggests that even for very hard problems, biasing against verbosity can be valuable.
>
> In this context, filtering based on conciseness is a practical way to avoid reinforcing these failure patterns. Empirically, Shortest-k GFPO works well because many long traces are unproductively verbose.
>
> **Ultimately, our goal is to teach models to think more in the _right_ settings.** We explicitly address this goal through two GFPO variants designed to preserve useful verbosity:
>
> - **Token Efficiency GFPO** rewards long reasoning traces when they lead to high rewards, ensuring that beneficial verbosity is retained.
>
> - **Adaptive Difficulty GFPO** estimates question difficulty on-the-fly and relaxes filtering for harder questions, preserving longer chains exactly when deeper reasoning is likely to help. We see this in action as Adaptive Difficulty matches or exceeds GRPO accuracy on medium and very hard AIME-25 problems while reducing average response lengths by up to 60% (Figure 3).
>
> **Thus, GFPO does not suppress productive verbosity; it selectively filters unproductive verbosity while preserving and reinforcing the cases where extended reasoning is genuinely beneficial.** This balance is central to why the method maintains accuracy while substantially reducing response length across math, STEM, and coding benchmarks.
>
> We hope that our comments provide added clarity on our method and address reviewer 3LZ1's questions. We would be happy to answer any other questions or comments you may have. Thank you!
>
> **References:**
>
> Vidhisha Balachandran, Jingya Chen, Lingjiao Chen, Shivam Garg, Neel Joshi, Yash Lara, John Langford, Besmira Nushi, Vibhav Vineet, Yue Wu, and Safoora Yousefi. Inference-time scaling for complex tasks: Where we stand and what lies ahead. 2025.
>
> Marah Abdin, Sahaj Agarwal, Ahmed Awadallah, Vidhisha Balachandran, Harkirat Behl, Lingjiao Chen, Gustavo de Rosa, Suriya Gunasekar, Mojan Javaheripi, Neel Joshi, Piero Kauffmann, Yash Lara, Caio César Teodoro Mendes, Arindam Mitra, Besmira Nushi, Dimitris Papailiopoulos, Olli Saarikivi, Shital Shah, Vaishnavi Shrivastava, Vibhav Vineet, Yue Wu, Safoora Yousefi, and Guoqing Zheng. Phi-4-reasoning technical report, 2025.

---

### Official Review · Reviewer_9jp4 · 2025-10-30

**Soundness:** 3
**Presentation:** 3
**Contribution:** 3
**Rating:** 6
**Confidence:** 4

**Summary:**

GFPO (Group Filtered Policy Optimization), which curbs this length explosion by sampling larger groups per problem and only training on responses filtered by length and token efficiency (reward per token). This can make LLM generate shorter reasoning traces during inference.

**Strengths:**

1. The intuition of sampling more and using more computational resources on hard questions is straightforward and reasonable.
2. The experiment shows the effectiveness of GFPO in different tasks, including mathematical, STEM, and coding reasoning.
3. The method itself is simple and easy to plug into any other RL post-training framworks.

**Weaknesses:**

1. The evaluated model size and model family are limited, which only contain the 14B Phi-4 model.
2. Considering this method is based on GRPO, maybe an analysis of training stability is needed.

**Questions:**

1. Is it possible to add some experiments on other sizes of models, besides 14 B?
2. Is it possible to add some experiments on other model families, such as Qwen, DeepSeek, and Llama?
3. Is it possible to include some analysis about the training stability?

---

> ### Author Response · Authors · 2025-11-24
> **Official Comment by Authors**
>
> We thank reviewer 9jp4 for their thoughtful review and feedback. We appreciate that they find GFPO to be a straightforward and reasonable method that is effective across different tasks and easy to integrate with different RL frameworks. Reviewer 9jp4 requests experiments on other model sizes (besides 14B), experiments on other model families beyond Phi, and some analysis on training stability. We have added new experiments and discussion to address each of these questions, which we summarize below. Please let us know if you have additional questions or feedback for us during the discussion period. Thank you!
>
> **“Is it possible to add some experiments on other sizes of models, besides 14B?”**
>
> To address this question, **we evaluate GFPO on the DeepSeek-R1-Distill-Qwen-7B and DeepSeek-R1-Distill-Llama-8B models**. DeepSeek-R1-Distill-Qwen-7B and DeepSeek-R1-Distill-Llama-8B are state-of-the-art models obtained by supervised fine-tuning of Qwen2.5-Math-7B and Llama-3.1-8B on DeepSeek-R1 reasoning traces. We train these models with both GRPO (group size=8) and 8/16 shortest-k GFPO, where we sample 16 responses per question but only train on the shortest 8.
>
> **Training setup:** We train with verl on 16 H100s with global batch size of 64 for 100 steps. We use Adam with learning rate=1e-6, cosine warmup for 10 steps, sampling temperature of T=1.0, KL regularization ($\beta=0.001$), and entropy coefficient ($\gamma=0.001$). We train with 16k context and reserve 1k tokens for the prompt.
>
> **Evaluation setup:** We compare the resulting RLed models with the base DeepSeek-R1-Distill-Qwen-7B and DeepSeek-R1-Distill-Llama-8B (denoted as SFT) models across AIME 25 (with 32 samples), AIME 25 (with 32 samples), and GPQA (with 5 samples). Note that our RL training set focuses on mathematical reasoning, so we evaluate GPQA as an OOD task to verify that RL training does not harm performance. Responses are sampled at T=0.6, top-p=0.95 as recommended for the Deepseek Distill Qwen Models, with 16k max token length.
>
> As in our other experiments, we report pass@1 accuracy (Acc), raw response length (Avg Rsp Len), and excess length reduction (% Len Inf ($\downarrow$)), which captures the fraction of length increase of GRPO eliminated by GFPO.
>
> ### DeepSeek-R1-Distill-Qwen-7B Results
> | Method   | AIME 25         |   |          |    AIME 24      |   |          | GPQA   |         |          |
> |----------|----------|----------|----------|----------|----------|----------|----------|----------|----------|
> |          | Acc      |Avg Rsp Len| % Len Inf ($\downarrow$)|Acc      |Avg Rsp Len| % Len Inf ($\downarrow$) |Acc      |Avg Rsp Len| % Len Inf ($\downarrow$) |
> | SFT  | 37.0 | 11.4k  | N/A | 49.5 | 10.7k | N/A| 47.4 | 7.5K | N/A |
> | GRPO  | 40.2  | 13.3k | 0 | 51.5 | 11.9k | 0 | 48.4 | 8.3k | 0 |
> | 8 of 16 GFPO | 39.1 | 11.9k | 63.0 | 52.4 | 11k | 39.6 | 48.0 | 7.9k | 48.9 |
>
> **For DeepSeek-R1-Distill-Qwen-7B, we find that GFPO reduces excess response length inflation by **63%**, **39.6%**, and **48.9%** on AIME 25, 24, and GPQA** while closely matching the accuracy of GRPO on each benchmark.
>
> ### DeepSeek-R1-Distill-Llama-8B Results
> | Method   | AIME 25         |   |          |    AIME 24      |   |          | GPQA   |         |          |
> |----------|----------|----------|----------|----------|----------|----------|----------|----------|----------|
> |          | Acc      |Avg Rsp Len| % Len Inf ($\downarrow$)|Acc      |Avg Rsp Len| % Len Inf ($\downarrow$) |Acc      |Avg Rsp Len| % Len Inf ($\downarrow$) |
> | SFT  | 29.8 | 12.7k  | N/A | 44.3 | 12.1k | N/A| 45.7 | 7.5k | N/A |
> | GRPO  | 33.0  | 10.4k | 0 | 47.9 | 10.3k | 0 | 45.5 | 7.6k | 0 |
> | 8 of 16 GFPO | 33.3 | 10.4k | Undef | 48.5 | 10.0k | Undef | 45.7 | 7.3k | 319 |
>
> For AIME 25 and AIME 24, we find that both GRPO and GFPO improve accuracy relative to the DeepSeek-R1-Distill-Llama-8B (SFT) model while also reducing response length. Since GRPO does not inflate length on these tasks, we do not report excess-length reduction values. **GFPO, however, consistently matches or further shortens GRPO's response lengths.**
>
> **For GPQA, GRPO incurs mild length inflation, whereas GFPO achieves a substantial length reduction**—producing responses even shorter than the SFT model while preserving accuracy.  This demonstrates that GFPO remains effective at curbing response length inflation while maintaining accuracy even on smaller models.
>
> We have updated Section 4.7 to include these results.

---

> ### Author Response · Authors · 2025-11-24
> **Official Comment by Authors**
>
> **“Is it possible to add some experiments on other model families, such as Qwen, DeepSeek, and Llama?”**
>
> The results above already evaluate GFPO on two new model families (Qwen and Llama). To further test whether our findings transfer to another model of the same size as Phi-4-Reasoning (14B) but in a different family, **we also evaluate GFPO on DeepSeek-R1-Distill-Qwen-14B**. This model is obtained by supervised fine-tuning of Qwen2.5-14B on DeepSeek-R1 reasoning traces. Following the same setup as for the 7B model, we train this model with GRPO and shortest 8/16 GFPO.
>
> **Training setup:** We follow the same training setup as for our Phi-4-reasoning model. We train with verl on 32 H100s with global batch size of 64 for 100 steps. We use Adam with learning rate=1e-7, cosine warmup for 10 steps, sampling temperature of T=1.0, KL regularization ($\beta=0.001$), and entropy coefficient ($\gamma=0.001$). We train with 32k context and reserve 1k tokens for the prompt.
>
> We evaluate on AIME 25 (32 samples), AIME 24 (32 samples), and GPQA (5 samples) with T=0.6, top-p=0.95 with 32k max token length and report pass@1 accuracy (Acc), raw response length (Avg Rsp Len), and excess length reduction (% Len Inf ($\downarrow$)).
>
> ### DeepSeek-R1-Distill-Qwen-14B Results
> | Method   | AIME 25         |   |          |    AIME 24      |   |          | GPQA   |         |          |
> |----------|----------|----------|----------|----------|----------|----------|----------|----------|----------|
> |          | Acc      |Avg Rsp Len| % Len Inf ($\downarrow$)|Acc      |Avg Rsp Len| % Len Inf ($\downarrow$) |Acc      |Avg Rsp Len| % Len Inf ($\downarrow$) |
> | SFT  | 48.2 | 12.3k  | N/A | 64.0 | 10.8k | N/A| 56.4 | 7k | N/A |
> | GRPO  | 51.9  | 16.1k | 0 | 69.3 | 13.9k | 0 | 56.8 | 8.8k | 0 |
> | 8 of 16 GFPO | 52.0 | 13.7k | 61.9 | 68.5 | 12.5k | 44.9 | 57.4 | 8.4k | 22.8 |
>
> **For DeepSeek-R1-Distill-Qwen-14B, we observe that GFPO reduces excess response length by **61.9%**, **44.9%**, and **22.8%** for AIME 25, AIME 24, and GPQA** compared to GRPO, while matching GRPO's accuracy. This further demonstrates that GFPO is effective across model families. We include these results in Section 4.7.

---

> ### Author Response · Authors · 2025-11-24
> **Official Comment by Authors**
>
> **"Is it possible to include some analysis about the training stability?"**
>
> Reviewer 9jp4's request for an analysis of training stability suggests that they are interested in whether GFPO causes unstable reward oscillations, KL spikes or collapses, or other signs of divergent training.
>
> To investigate this, **we have added training curves for KL loss, reward, average response length, and AIME 2025 accuracy (pass@1 averaged over 8 samples) as a function of training step, comparing GRPO and our primary GFPO variants** (Shortest 8/16, Shortest 8/24, Token Efficiency GFPO, and Adaptive Difficulty GFPO) in Figure 8. We also include training curves comparing the training stability of GFPO with Dr. GRPO in Figure 9.
>
> **Empirically, we observe no sudden KL explosions or collapses during GFPO training, and GFPO’s KL curves are as smooth or smoother than GRPO’s.** All methods show some reward oscillation during training, but all converge monotonically toward the same reward. Throughout training, GFPO continues to substantially suppress response-length growth. We also find that GFPO typically achieves its peak accuracy earlier than GRPO, while matching or improving on GRPO’s best accuracy in almost all cases (e.g., Shortest 8/24, Token Efficiency, and Adaptive Difficulty GFPO) (Figure 8).
>
> **Meanwhile, we find training with Dr. GRPO, a prominent alternative method for efficient reasoning, to be relatively unstable**—with occasional spikes in gradient norm, KL divergence, entropy, and response length. These arise when a batch contains a few very long trajectories with large (unnormalized) advantages. Because Dr. GRPO drops both std normalization and per-response length normalization, such trajectories dominate the update and cause transient but sharp jumps in the PPO statistics which can cause training to diverge (Figure 9).
>
> **Taken together, these results indicate that GFPO training is as stable as GRPO.**
>
> We have included our results on the DeepSeek-R1-Distill-Qwen-7B and 14B models into Section 4.6 and included our discussion on training stability in Appendix A.7. We hope that our added experiments and analysis address reviewer 9jp4's questions and concerns. We appreciate your feedback and are happy to answer any further questions.

---

### Official Review · Reviewer_rWEa · 2025-11-14

**Soundness:** 4
**Presentation:** 4
**Contribution:** 3
**Rating:** 8
**Confidence:** 3

**Summary:**

The authors propose GFPO (group filtered policy optimization), a method to encourage concise responses in reasoning models by using both token efficiency and length as filters to select the best examples over a larger set of rollouts. Through this larger amount of oversampling they find better examples to optimize for token efficiency. It adds 7% to the training time but saves 90 seconds on average over long queries.

The adaptive version dynamically scales the number of samples taken from the model based on an estimate of the difficulty. If rollouts continue to fail, more are sampled until a maximum. There are a few insights guiding this: easy questions don’t need as much reinforcement, so we can stick to just giving a few short demonstrations. Harder questions may have different ways of expressing them. The authors claim theirs is the first RLVR method to do something like this.

They test using GFPO to RLVR Phi-4-reasoning against a baseline that used GRPO with 72k math problems. Their approach uses the same data setup as Phi-4-reasoning-plus, so it’s a fair comparison. All parameters are kept the same. GFPO delivers slightly better accuracy and significant improvements in length over GRPO.

I’m not really the right person to review this as I don’t work on RL, so this is a low confidence review.

**Strengths:**

- Simple, elegant, but novel (I think) idea
- High-validity experiment by testing against an existing baseline with everything else held equal
- Clear demonstration of improvement in performance
- Lengthy and detailed analysis

**Weaknesses:**

I am pretty convinced by the results, but it would be nice to see the experiment replicated for at least one other model so we can know it’s not a fluke.

**Questions:**

N/a, sorry my review is late

---

> ### Author Response · Authors · 2025-11-25
> **Official Comment by Authors**
>
> We thank reviewer rWEa for their thoughtful review of our work. We appreciate that they find GFPO to be simple, elegant, and novel and find our analysis to be thorough. Reviewer rWEa is satisfied with our current results but requests GFPO to be validated on at least one other model. To address this question, we run additional experiments on new models and summarize our results below. Please let us know if you have any other questions or feedback during the discussion period. Thank you!
>
> **"it would be nice to see the experiment replicated for at least one other model"**
>
> To address this suggestion, **we additionally evaluate GFPO on the DeepSeek-R1-Distill-Qwen-7B and DeepSeek-R1-Distill-Qwen-14B models** to determine if GFPO generalizes to other model sizes (beyond 14B) and model families (beyond Phi). DeepSeek-R1-Distill-Qwen-7B is a state-of-the-art 7B model obtained by supervised fine-tuning of Qwen2.5-Math-7B on DeepSeek-R1 reasoning traces. DeepSeek-R1-Distill-Qwen-14B is a strong alternative to the Phi-4-reasoning-plus model derived by supervised fine-tuning of Qwen2.5-14B on DeepSeek-R1 samples.
>
> We train these models with both GRPO (group size=8) and 8/16 shortest-k GFPO, where we sample 16 responses per question but only train on the shortest 8.
>
> **Training setup:** For the 7B model, we train with verl on 16 H100s with global batch size of 64 for 100 steps. We use Adam with learning rate=1e-6, cosine warmup for 10 steps, sampling temperature of T=1.0, KL regularization ($\beta=0.001$), and entropy coefficient ($\gamma=0.001$). We train with 16k context and reserve 1k tokens for the prompt.
> For the 14B model, we match the training setup of GFPO on the Phi-4-reasoning model and train with 32 H100s, with learning rate=1e-7, and 32k context. Other hyperparameters match the 7B setting.
>
> **Evaluation Setup:** We compare the resulting RLed models with the base DeepSeek-R1-Distill-Qwen-7B and DeepSeek-R1-Distill-Qwen-14B models (denoted as SFT) across AIME 25 (with 32 samples), AIME 25 (with 32 samples), and GPQA (with 5 samples). Note that our RL training set focuses on mathematical reasoning, so we evaluate GPQA as an OOD task to verify that RL training does not harm performance. Responses are sampled at T=0.6, top-p=0.95 as recommended for the Deepseek Distill Qwen Models, with 16k max token length for the 7B model and 32k max token length for the 14B model.
>
> As in our prior experiments, we report pass@1 accuracy, raw response length, and excess length reduction, which captures the fraction of length increase of GRPO eliminated by GFPO.
>
> ### DeepSeek-R1-Distill-Qwen-7B Results
> | Method   | AIME 25         |   |          |    AIME 24      |   |          | GPQA   |         |          |
> |----------|----------|----------|----------|----------|----------|----------|----------|----------|----------|
> |          | Acc      |Avg Rsp Len| % Len Inf ($\downarrow$)|Acc      |Avg Rsp Len| % Len Inf ($\downarrow$) |Acc      |Avg Rsp Len| % Len Inf ($\downarrow$) |
> | SFT  | 37.0 | 11.4k  | N/A | 49.5 | 10.7k | N/A| 47.4 | 7.5K | N/A |
> | GRPO  | 40.2  | 13.3k | 0 | 51.5 | 11.9k | 0 | 48.4 | 8.3k | 0 |
> | 8 of 16 GFPO | 39.1 | 11.9k | 63.0 | 52.4 | 11k | 39.6 | 48.0 | 7.9k | 48.9 |
>
> **For DeepSeek-R1-Distill-Qwen-7B, we find that GFPO reduces excess response length inflation by **63%**, **39.6%**, and **48.9%** on AIME 25, 24, and GPQA** while closely matching the accuracy of GRPO on each benchmark. This demonstrates that GFPO remains effective at curbing response length inflation while maintaining accuracy even on smaller models.
>
> ### DeepSeek-R1-Distill-Qwen-14B Results
> | Method   | AIME 25         |   |          |    AIME 24      |   |          | GPQA   |         |          |
> |----------|----------|----------|----------|----------|----------|----------|----------|----------|----------|
> |          | Acc      |Avg Rsp Len| % Len Inf ($\downarrow$)|Acc      |Avg Rsp Len| % Len Inf ($\downarrow$) |Acc      |Avg Rsp Len| % Len Inf ($\downarrow$) |
> | SFT  | 48.2 | 12.3k  | N/A | 64.0 | 10.8k | N/A| 56.4 | 7k | N/A |
> | GRPO  | 51.9  | 16.1k | 0 | 69.3 | 13.9k | 0 | 56.8 | 8.8k | 0 |
> | 8 of 16 GFPO | 52.0 | 13.7k | 61.9 | 68.5 | 12.5k | 44.9 | 57.4 | 8.4k | 22.8 |
>
> **For DeepSeek-R1-Distill-Qwen-14B, we observe that GFPO reduces excess response length by **61.9%**, **44.9%**, and **22.8%** for AIME 25, AIME 24, and GPQA** compared to GRPO, while matching GRPO's accuracy. This further demonstrates that GFPO is effective across model families.
>
> We have updated Section 4.7 to include these results. We hope these experiments address any reservations reviewer rWEa has about our work. We are happy to engage in further discussions to resolve any remaining questions or comments. Thank you!

---

> ### Author Response · Authors · 2025-12-03
> **Official Comment by Authors**
>
> We additionally evaluate GFPO on **DeepSeek-R1-Distill-Llama-8B**, a state-of-the-art 8B model obtained by fine-tuning Llama-3.1-8B on DeepSeek-R1 reasoning traces. We use the same training and evaluation setup as in our DeepSeek-R1-Distill-Qwen-7B experiments. The SFT model corresponds to DeepSeek-R1-Distill-Llama-8B without any RL tuning.
>
> ### DeepSeek-R1-Distill-Llama-8B Results
> | Method   | AIME 25         |   |          |    AIME 24      |   |          | GPQA   |         |          |
> |----------|----------|----------|----------|----------|----------|----------|----------|----------|----------|
> |          | Acc      |Avg Rsp Len| % Len Inf ($\downarrow$)|Acc      |Avg Rsp Len| % Len Inf ($\downarrow$) |Acc      |Avg Rsp Len| % Len Inf ($\downarrow$) |
> | SFT  | 29.8 | 12.7k  | N/A | 44.3 | 12.1k | N/A| 45.7 | 7.5k | N/A |
> | GRPO  | 33.0  | 10.4k | 0 | 47.9 | 10.3k | 0 | 45.5 | 7.6k | 0 |
> | 8 of 16 GFPO | 33.3 | 10.4k | Undef | 48.5 | 10.0k | Undef | 45.7 | 7.3k | 319 |
>
> For AIME 25 and AIME 24, we find that both GRPO and GFPO improve accuracy relative to the SFT model while also reducing response length. Since GRPO does not inflate length on these tasks, we do not report excess-length reduction values. **GFPO, however, consistently matches or further shortens GRPO's response lengths.**
>
> **For GPQA, GRPO incurs mild length inflation, whereas GFPO achieves a substantial length reduction**—producing responses even shorter than the SFT model while preserving accuracy. Overall, GFPO generalizes well to this model family, improving response efficiency while maintaining GRPO-level accuracy. We have updated Section 4.7 to include these results.

---

### Author Response · Authors · 2025-12-03
**Author Response Summary**

**We thank all reviewers for their thoughtful and constructive feedback.** We appreciate reviewers highlighting that GFPO is simple and elegant, generalizes well across settings, and offers a rare combination of improved inference efficiency without loss of accuracy.

We have responded to each reviewer individually and believe we have addressed all raised concerns. Based on reviewer comments, we have strengthened the submission with new experiments, analyses, and clarifications (changes highlighted in blue). Below we summarize the key updates:

- **Expanded Model Coverage (Section 4.7).** We add experiments on DeepSeek-R1-Distill-Qwen-7B, DeepSeek-R1-Distill-Llama-8B, and DeepSeek-R1-Distill-Qwen-14B, in addition to our original Phi-4-reasoning results, demonstrating that GFPO generalizes across multiple model families and sizes. *(Reviewers rWEa, 9jp4, g7JB)*

- **Comparison to Dr. GRPO (Section 4.6 and Appendix A.7).** We compare GFPO to Dr. GRPO, a prominent alternative for token-efficient reasoning, and find that GFPO consistently delivers higher accuracy and more token-efficient responses across tasks. *(Reviewer g7JB)*

- **Training Stability Analysis (Appendix A.7).** We add analysis showing that GFPO matches GRPO’s training stability, while Dr. GRPO can exhibit occasional instabilities. *(Reviewers 9jp4, 3LZ1)*

- **Guidance on Tuning and Scaling k/G (Appendix A.5).** We provide practical guidelines for practitioners on tuning and scaling the retention ratio k/G, grounded in empirical findings. *(Reviewers 3LZ1, g7JB)*

- **Clarifications on Adaptive Difficulty GFPO (Section 2.1 and Appendix A.6).** We expand the discussion of how we mitigate early-stage noise and include additional notes on our momentum-based explorations. *(Reviewer 3LZ1)*

Thank you for your consideration!

---

### Meta-Review · Area_Chair_Go5X · 2025-12-09

**Summary:**

3/4 reviewers recommended acceptance in their initial reviews. The common concern was generalizability to other models.

**Reviewer Concerns:**

Generalizability to other models
- Very well addressed by the addition of DeepSeek-R1-Distill-Qwen-7B, DeepSeek-R1-Distill-Llama-8B, and DeepSeek-R1-Distill-Qwen-14B

Reviewer 3LZ1 concerns
- The reviewer had doubts about the efficacy of the proposed method's design choices. I think the new experiments show that the proposed method mostly outperforms the baselines, so I would consider this solved.

Reviewer g7JB concerns
- The reviewer was concerned about the number of baseline methods, which was only partly addressed by including 1 additional baseline.

**Reviewer Scores:**

9jp4, 3LZ1 would both raise their scores, as their concerns are either about, or can be resolved by, conducting additional experiments on more models.

rWEa didn't have any major concerns, and g7JB's concern on including more baseline methods was only partly addressed addressed. They would not have changed their scores.

---

### Decision · Program_Chairs · 2026-01-26

Accept (Poster)